# Dynamic changes in carbonate chemistry in the microenvironment around single marine phytoplankton cells

Abdul Chrachri [1], Brian M. Hopkinson[2], Kevin Flynn [3], Colin Brownlee[1,4] & Glen L. Wheeler[1]

Photosynthesis by marine diatoms plays a major role in the global carbon cycle, although the precise mechanisms of dissolved inorganic carbon (DIC) uptake remain unclear. A lack of direct measurements of carbonate chemistry at the cell surface has led to uncertainty over the underlying membrane transport processes and the role of external carbonic anhydrase (eCA). Here we identify rapid and substantial photosynthesis-driven increases in pH and $[CO_3^{2-}]$ primarily due to the activity of eCA at the cell surface of the large diatom *Odontella sinensis* using direct simultaneous microelectrode measurements of pH and $CO_3^{2-}$ along with modelling of cell surface inorganic carbonate chemistry. Our results show that eCA acts to maintain cell surface $CO_2$ concentrations, making a major contribution to DIC supply in *O. sinensis*. Carbonate chemistry at the cell surface is therefore highly dynamic and strongly dependent on cell size, morphology and the carbonate chemistry of the bulk seawater.

[1] Marine Biological Association, Plymouth PL1 2PB, UK. [2] Department of Marine Sciences, University of Georgia, Athens, 30602-3636 GA, USA. [3] Biosciences, Swansea University, Singleton Park, Swansea SA2 8PP, UK. [4] School of Ocean and Earth Science, University of Southampton, Southampton SO14 3ZH, UK. Correspondence and requests for materials should be addressed to C.B. (email: cbr@mba.ac.uk) or to G.L.W. (email: glw@mba.ac.uk)

The diatoms are an abundant group of marine phytoplankton that contribute as much as 40% of marine primary productivity[1]. Despite the importance of diatoms in global carbon cycling, significant uncertainty remains around their mechanisms of uptake for dissolved inorganic carbon (DIC) in support of photosynthesis. Like most other marine phytoplankton, diatoms need to operate a carbon concentrating mechanism (CCM) due to the low availability of $CO_2$ (the substrate for carbon fixation by the enzyme RuBisCO) in the alkaline pH of seawater, with <1% of DIC present as $CO_2$ in present day oceans[2]. The low availability of $CO_2$ in seawater is problematic because RuBisCO has a relatively low affinity and specificity for $CO_2$, requiring the cell to increase the concentration of $CO_2$ at the site of fixation to minimise the alternative reaction with $O_2$[3]. Marine diatoms have therefore evolved a variety of biophysical mechanisms to improve the supply of DIC to the cell surface and to concentrate $CO_2$ around RuBisCO[4,5]. In addition, some marine diatoms operate a biochemical CCM, in which $CO_2$ is initially fixed into C4 organic acids in the cytosol and later released at the site of RuBisCO, although the exact nature of single-cell C4 photosynthesis in diatoms has been extensively debated[6–8].

The supply of DIC to the cell surface is a critical aspect of the CCM, as it is affected significantly by cell size. Diatoms are capable of both $CO_2$ and $HCO_3^-$ uptake across the plasma membrane[9,10]. The plasma membrane is relatively permeable to $CO_2$ allowing uptake to occur via passive diffusion, whereas transport of the much more abundant $HCO_3^-$ must be facilitated by dedicated transporters. Diatom genomes possess several potential $HCO_3^-$ transporters belonging to the SLC4 and SLC26 families of transporters. Characterisation of PtSLC4-2 from *Phaeodactylum tricornutum* indicated that it contributed significantly to $Na^+$-coupled $HCO_3^-$ transport during photosynthesis[11]. In contrast to active transport processes, diffusive uptake of $CO_2$ can only occur if the cell is able to maintain an inward gradient for $CO_2$ across the plasma membrane. For a cell relying on $CO_2$ uptake, modelling studies indicate that <5% of the $CO_2$ at the cell surface is likely to be supplied by conversion of $HCO_3^-$ to $CO_2$, due the slow rate of the uncatalysed reaction[12]. $CO_2$ supply at the cell surface is therefore limited by diffusion and maintaining an inward $CO_2$ gradient across the plasma membrane is a much greater problem for large cells that have a significant diffusive boundary layer[12–14]. Large cells may overcome this diffusive limitation either by direct uptake of $HCO_3^-$ or by using the enzyme external carbonic anhydrase (eCA) to increase the supply of $CO_2$ at the cell surface. It is likely that many species employ both mechanisms, although the role of eCA in photosynthetic DIC uptake in marine diatoms has been much debated[15,16].

Improved knowledge of these cellular mechanisms is critical for our understanding of the response of diatom communities to predicted future changes in ocean carbonate chemistry. For example, experimental analyses have demonstrated that growth at elevated $CO_2$ increases the growth rate of large diatoms by up to 30%, whereas the growth enhancement in smaller species was much more modest (<5%)[17]. The significant growth enhancement of large diatoms may be due to the increased diffusive supply of $CO_2$ and/or a decreased metabolic investment in the CCM components[17]. Future changes in ocean carbonate chemistry may therefore lead to shifts in the size and productivity of diatom communities that will have an important implication on global carbon cycling through their influence on the rates of carbon export from the surface ocean.

It was initially assumed that the primary role of eCA in marine diatoms and other algae is to catalyse the conversion of $HCO_3^-$ to $CO_2$ at the cell surface[18–20]. eCA would therefore be expected to be more important in larger diatom species. A survey of 17 marine diatoms indicated that there is considerable diversity in the presence of eCA activity between different species, but found no correlation between eCA activity and the relative C demand: supply of each species[21]. eCA is present in most centric diatoms, although in smaller species it is only induced and required at very low DIC concentrations[15,22]. Although no overall relationship was found between the contribution of eCA to photosynthesis and cell size, larger centric diatom species exhibit a requirement for eCA at ambient DIC concentrations, lending some support to the increased requirement for eCA in larger cells[23]. Hopkinson et al.[15] proposed that even relatively small increases in diffusive $CO_2$ supply due to eCA are likely to increase the efficiency of the CCM.

Other lines of evidence suggest that the primary role of eCA is not to increase the supply of $CO_2$ at the cell surface. Studies across a range of diatom species using the isotope disequilibrium technique to discriminate between $CO_2$ and $HCO_3^-$ uptake surprisingly revealed a positive correlation between eCA activity and the proportion of DIC taken up across the plasma membrane as $HCO_3^-$ ($fHCO_3^-$)[21]. Similar results in other marine diatoms have been observed using membrane-inlet mass spectrometry (MIMS)[9,16,24]. These correlations have led to proposals that the primary role of eCA in marine phytoplankton is actually to assist active $HCO_3^-$ uptake by scavenging $CO_2$ leaking out of the cell[16,21,24]. In this scenario, active $HCO_3^-$ transport may result in an elevated $[CO_2]$ inside the cell, which can diffuse out of the cell across the plasma membrane. The activity of eCA at the cell surface could minimise this diffusive loss of $CO_2$ by catalysing its conversion to $HCO_3^-$. Although there is a strong correlation between eCA activity and the proportion of DIC uptake as $HCO_3^-$, definitive evidence supporting this role for eCA is lacking. Other alternative roles for eCA include the regulation of cell surface pH, as proposed in mammalian cells[25,26].

The requirement and physiological role of eCA in photosynthetic DIC uptake in marine diatoms therefore remains unclear, despite the ecological importance of carbon assimilation by this group. The different experimental approaches used may have contributed to this uncertainty, as some of these may underestimate the contribution of eCA[27,28]. Both membrane-inlet mass spectrometry (MIMS) and the isotope disequilibrium technique require the use of mathematical models to interpret the findings and therefore incorporate a number of assumptions relating to carbonate chemistry within the cell surface boundary layer[15]. Whilst the impact of cell size and morphology on the acquisition of DIC has been extensively modelled[12], direct measurements of the diffusive boundary layer surrounding phytoplankton cells are limited. Measurements of the micro-environment of multicellular organisms[29,30] or very large single cells (such as foraminifera, which contain photosynthetic endosymbionts)[31] demonstrate that photosynthetic DIC uptake is often associated with a significant rise in pH. Previous microelectrode measurements of cell surface pH in the large diatom *Coscinodiscus wailesii* indicate that diatom cells may also experience significant changes in pH, although the underlying processes have not been explored[32]. Measurements using pH-responsive fluorescent dyes have also demonstrated significant light-dependent increases in cell surface pH in diatoms[33]. Photosynthetic DIC uptake could theoretically contribute to the light-dependent increases in cell surface pH in diatoms through a number of mechanisms; drawdown of $CO_2$, conversion of $HCO_3^-$ to $CO_2$ at the cell surface or uptake of $HCO_3^-$ accompanied by uptake of $H^+$ or extrusion of a base ($OH^-$)[33]. Clearly, better definition of carbonate chemistry in the

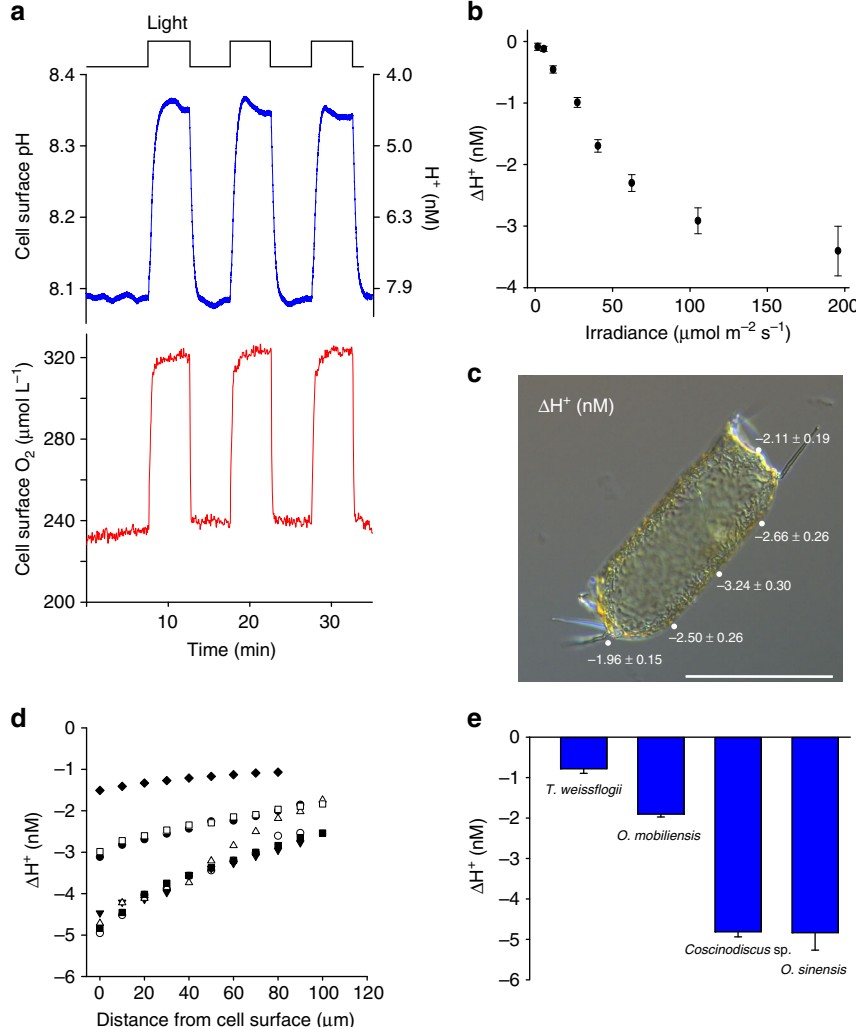

**Fig. 1** Photosynthetic DIC uptake results in an increase in pH at the cell surface. **a** Light-dependent changes in cell surface pH (upper) and [O$_2$] (lower) around the large diatom *Odontella sinensis* measured using microelectrodes. Upon illumination there are very rapid increases in pH and [O$_2$]. **b** The increase in cell surface pH is strongly dependent on irradiance. The mean change in cell surface [H$^+$] ($\pm$s.e.m.) following illumination is shown ($n = 12$ cells). **c** Brightfield microscopy image of an *O. sinensis* cell illustrating spatial variability in cell surface pH in the light. The mean change in [H$^+$] ($\pm$s.e.m.) following illumination was measured at different positions around the cell in seawater at pH 8.0 ($n = 12$ cells). The positions of the microelectrode are shown around a representative cell. Bar = 100 μm. **d** The zone of elevated pH extends significantly away from the cell. For each illuminated cell, pH was recorded at 10 μm increments away from the cell. The change in [H$^+$] from the bulk seawater is shown (pH 8.0). $n = 7$ cells. **e** Comparison of the light-dependent increase in cell surface pH in four centric diatoms; *Thalassiosira weissflogii* (approximate length 20–25 μm), *Odontella mobiliensis* (length 40–60 μm), *Coscinodiscus* sp. (diameter 140–170 μm) and *O. sinensis* (length 150–250 μm). The mean light-dependent change in cell surface [H$^+$] in seawater at pH 8.0 is shown ($\pm$s.e.m.) ($n = 12$)

microenvironment is required to understand the relative contribution of these processes to photosynthetic DIC uptake.

In order to better define the mechanisms of photosynthetic DIC uptake and the roles of eCA in this process, we set out to examine the key predictions from cellular modelling studies relating to diffusive limitation of CO$_2$ supply to large phytoplankton[12]. We report direct measurements of the carbonate chemistry in the microenvironment around single cells of *Odontella sinensis*, a cosmopolitan large centric diatom that is common in European coastal waters[34,35]. Using ion-selective microelectrodes and selective inhibitors of eCA, we demonstrate that eCA activity plays a major role in photosynthetic DIC uptake in *O. sinensis* and that eCA activity is primarily responsible for the light-dependent perturbations in carbonate chemistry at the cell surface. By integrating our findings with cellular models, we conclude that the primary role of eCA in marine diatoms is to increase the supply of CO$_2$ to the cell surface by catalysing its conversion from HCO$_3^-$ and that the properties of

eCA enable the cell to counter the impacts of diffusive limitation imposed by cell size and morphology.

## Results

**Light-dependent fluctuations in pH at the cell surface**. To examine how processes within the boundary layer may influence photosynthetic DIC uptake, we positioned a pH microelectrode and an O$_2$ optode against the frustule of *O. sinensis*. In the dark, pH and [O$_2$] at the cell surface were similar to the bulk seawater (pH 8.0), with a small increase in cell surface [H$^+$] relative to the bulk seawater of $0.22 \pm 0.07$ nM and a small decrease in [O$_2$] ($98.53 \pm 0.57\%$ of the bulk [O$_2$], $n = 4$ cells). This indicates that in non-illuminated cells respiration and other metabolic processes only have a minor impact on the microenvironment around the cell. However, on application of incident white light for 300 s we observed a rapid and substantial increase in cell surface pH and

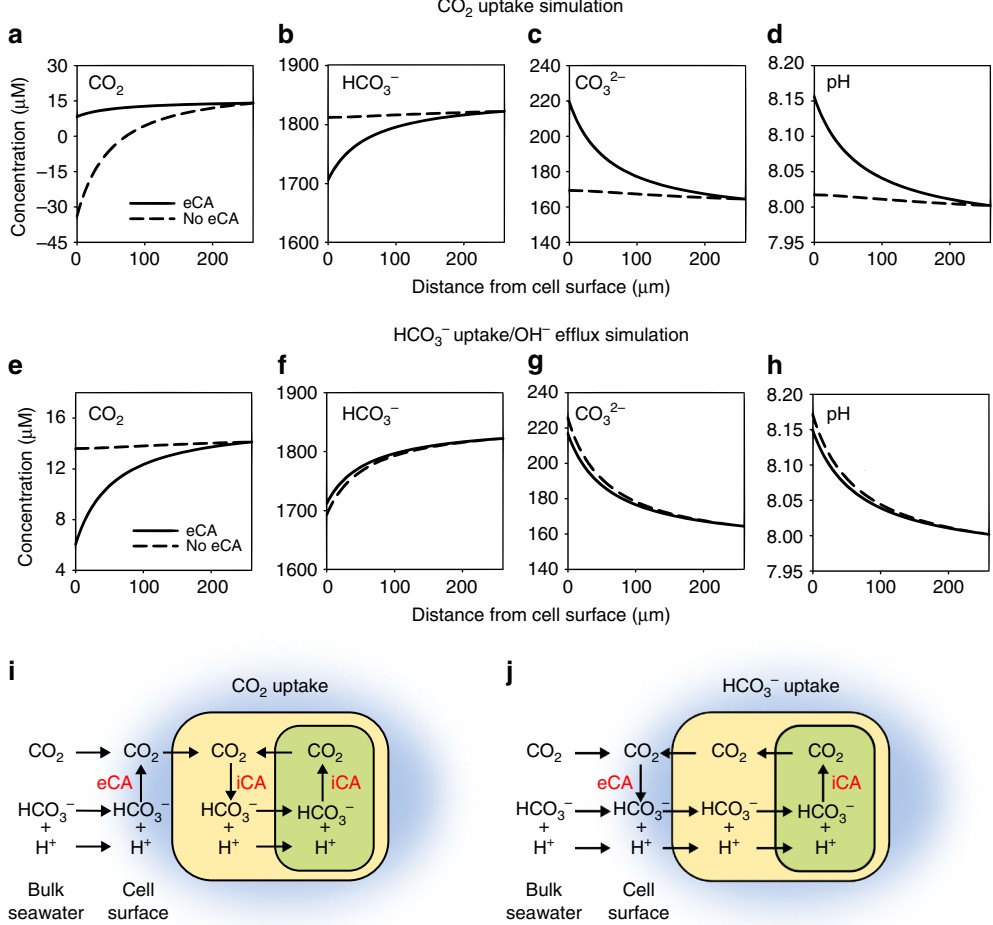

**Fig. 2** Cellular modelling of carbonate chemistry around a large diatom cell. Simulated profiles of inorganic carbon species ($CO_2$, $HCO_3^-$, $CO_3^{2-}$) and pH around a large ($r = 60\,\mu m$) photosynthesising cell. The horizontal axis represents distance away from the cell surface. **a–d** A cell taking up only $CO_2$ for photosynthesis in the presence (solid line) and absence (dashed-line) of extracellular carbonic anhydrase (eCA). **e–h** A cell taking up only $HCO_3^-$ for photosynthesis but exporting $OH^-$ to maintain internal pH and charge balance. The model shows that eCA is necessary to support substantial rates of $CO_2$ uptake and that surface pH is dependent on eCA activity only when $CO_2$ uptake occurs. Note that the model assumes a fixed rate of $CO_2$ uptake, which results in a negative value of $[CO_2]$ at the cell surface, illustrating that the combination of uncatalysed conversion from $HCO_3^-$ and diffusion is insufficient to supply $CO_2$ at this rate. The cell size approximates a typical *O. sinensis* cell and the eCA activity in the model is equivalent to that measured in *O. sinensis* ($8.3 \times 10^{-5}\,cm^3\,s^{-1}$). **i** Schematic illustration of the major DIC and $H^+$ fluxes during $CO_2$ uptake. eCA catalyses the conversion of $HCO_3^-$ to $CO_2$ at the cell surface (consuming $H^+$) to maintain the inward concentration gradient for $CO_2$. iCA intracellular carbonic anhydrase. **j** DIC and $H^+$ fluxes during active $HCO_3^-$ uptake. In this scenario, eCA could act to minimise $CO_2$ loss from the cell, converting $CO_2$ leaking across the plasma membrane to $HCO_3^-$ (generating $H^+$). For simplicity, the schematic shows $H^+$ uptake co-occurring with $HCO_3^-$ uptake, although $OH^-$ efflux was used in the cellular model described above

$[O_2]$ (Fig. 1a). The elevation in cell surface pH was strongly dependent on irradiance, with a mean decrease in cell surface $[H^+]$ of $3.40 \pm 0.4\,nM$ ($n = 12$) observed at a photosynthetic photon flux density (PPFD) of $200\,\mu mol\,m^{-2}\,s^{-1}$ (Fig. 1b). This equates to a cell surface pH of 8.18 relative to a bulk seawater pH of 8.0.

To examine the effect of cell morphology on the microenvironment, we placed a pH microelectrode at various positions along the length of an illuminated cell. We found that light-induced increases in cell surface pH were greatest in the central region, where diffusive limitation is predicted to be the greatest (Fig. 1c). In seawater with a bulk pH 8.0, the mean decrease in cell surface $[H^+]$ caused by illumination was $3.24 \pm 0.30\,nM$ at the centre of the cell, whereas it was $1.94 \pm 0.15\,nM$ at the tip of the cell ($n = 12$ cells). This indicates that the pH is not uniform at the cell surface of *O. sinensis* during photosynthetic DIC uptake, although the chloroplasts are evenly distributed along the length

of the cell (Supplementary Fig. 1). Moving the pH microelectrode away from the cell in $10\,\mu m$ increments (up to $100\,\mu m$) demonstrated that the zone of elevated pH extends significantly away from the cell (Fig. 1d). Buffering cell surface pH at 8.2 through the addition of $10\,mM$ HEPES almost completely inhibited the rise in cell surface pH ($\Delta H^+$ $13.9 \pm 1.1\%$ of control), but had little impact on the rate of photosynthetic $O_2$ evolution ($102.4 \pm 1.2\%$ of control, $n = 4$ cells) (Supplementary Fig. 2). Thus, the significant changes in cell surface pH experienced by *O. sinensis* do not appear to have a detrimental impact on cell physiology, suggesting that the cell can readily tolerate such pH changes.

Analysis of three further centric diatom species demonstrated that all exhibit a clear increase in cell surface pH upon illumination (Fig. 1e), with the greatest increase observed around the larger species, *Coscinodiscus* sp. and *O. sinensis*. Whilst direct comparisons between species must also consider differences in

morphology and photosynthetic rate, these measurements clearly illustrate that large diatoms commonly experience significant changes in pH within the boundary layer, supporting the predictions of previous cellular modelling studies[12,36].

**Modelling cell surface carbonate chemistry.** To better understand the cellular mechanisms that underlie the observed changes in cell surface pH, we employed a cellular modelling approach to simulate carbonate chemistry in the boundary layer around a diatom cell[15]. For simplicity, the model employs a spherical cell (radius 60 μm) and therefore does not explore the additional impacts of cellular morphology. The model was parameterised using an estimated photosynthetic rate and measured values of eCA activity in *O. sinensis* (Methods).

First, we examined direct $CO_2$ uptake in the presence or absence of eCA. The simulations clearly demonstrate that a large diatom cell, such as *O. sinensis* or *Coscinodiscus* sp., cannot rely on diffusive entry of $CO_2$ alone, as $CO_2$ taken up across the plasma membrane cannot be replaced sufficiently rapidly either by diffusion from the bulk seawater or by the uncatalysed reaction from $HCO_3^-$ at the surface (Fig. 2a). Note that the model assumes a fixed rate of DIC transport, although diffusive entry of $CO_2$ would not continue if the cell could not maintain an inward gradient[37]. The model shows that this diffusion limitation of $CO_2$ can be countered by the activity of eCA at the cell surface, enabling sufficiently rapid conversion of $HCO_3^-$ to $CO_2$ to support high rates of photosynthesis. The level of eCA activity has a dramatic impact on pH at the cell surface, with $[H^+]$ decreasing from 10 to 7 nM due to the demand for $H^+$ in the generation of $CO_2$ from $HCO_3^-$ by eCA (Fig. 2a–d). There is also an increase in

$[CO_3^{2-}]$ (170–210 μM) and a decrease in $[HCO_3^-]$ (1.82–1.72 mM) at the cell surface. These model simulations suggest that eCA activity is essential to sustain high rates of $CO_2$ uptake in large diatoms and that eCA has an important influence on the microenvironment around the cell.

We next simulated a cell using $HCO_3^-$ uptake for photosynthesis. $HCO_3^-$ transport was simulated with an equimolar $OH^-$ efflux to maintain charge balance during the intracellular generation of $CO_2$ from $HCO_3^-$ and $H^+$. The model indicates that $HCO_3^-$ uptake results in a decrease in $[HCO_3^-]$ at the cell surface, although the proportion of the $HCO_3^-$ pool that is depleted is minimal due to the much greater $[HCO_3^-]$ in seawater compared to $[CO_2]$. Thus, $HCO_3^-$ uptake is not subject to diffusive limitation (Fig. 2e–j). The $OH^-$ efflux contributes to a decrease in $[H^+]$ and an increase in $[CO_3^{2-}]$ at the cell surface, but there is little impact on $[CO_2]$. Carbonate chemistry in the boundary layer is therefore broadly similar to that observed during eCA-mediated $CO_2$ uptake. However, the important distinction is that during $CO_2$ uptake eCA activity defines the carbonate chemistry, whereas during $HCO_3^-$ uptake eCA has little impact on $[H^+]$, $[HCO_3^-]$ or $[CO_3^{2-}]$. The simulations of eCA activity during $HCO_3^-$ uptake indicate a depletion of $[CO_2]$ at the cell surface, as it facilitates $CO_2$ conversion to $HCO_3^-$.

**eCA activity contributes to increases in cell surface pH.** Our modelling studies indicate that the light-dependent elevations in cell surface pH observed around a large diatom cell are likely to be caused by either eCA-catalysed $CO_2$ generation or by $H^+$ uptake (or $OH^-$ efflux) in combination with $HCO_3^-$ uptake. We therefore examined whether *O. sinensis* demonstrates eCA activity using a

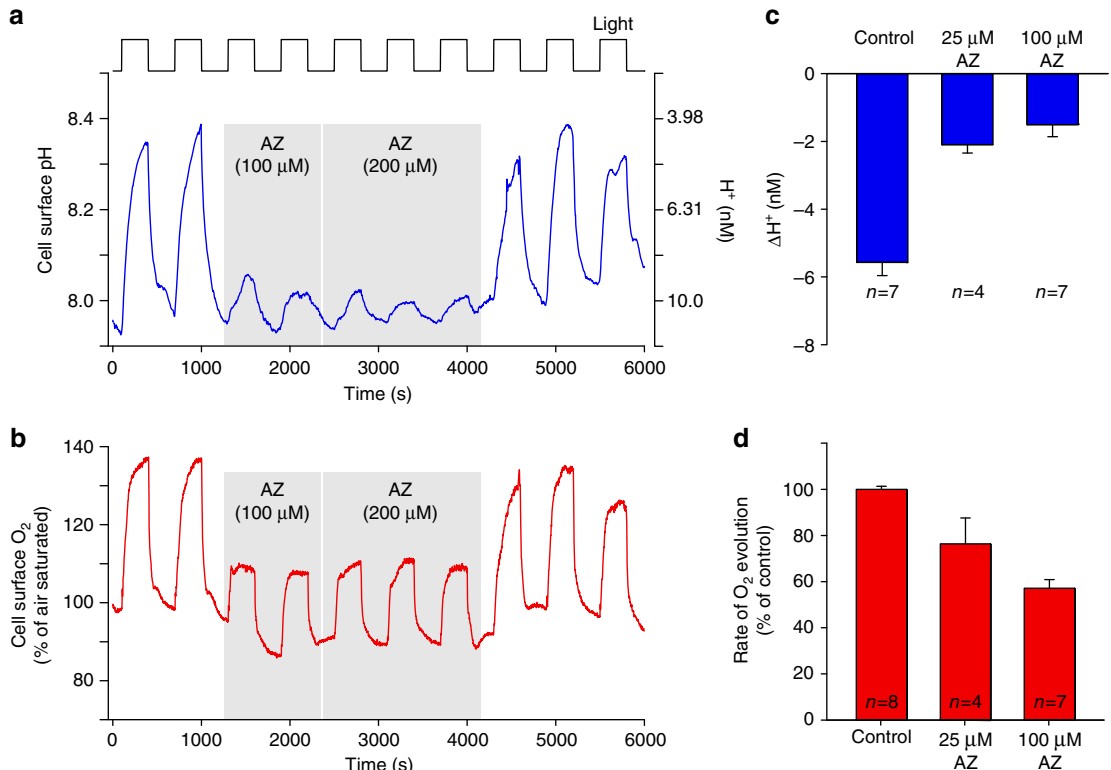

**Fig. 3** Changes in cell surface pH in *O. sinensis* due to the activity of external carbonic anhydrase. **a** Acetazolamide (AZ), an inhibitor of external carbonic anhydrase (eCA), has a significant impact on cell surface pH. Light-dependent increases in cell surface pH were measured using a pH microelectrode. The addition of AZ largely inhibits the increase in cell surface pH. **b** Light-dependent increases in $[O_2]$ for the cell shown in **a**. The addition of AZ substantially reduces the light-dependent increase in $[O_2]$ around the cell. The inhibitory effect of AZ is rapidly reversed. **c** Mean changes in cell surface $[H^+]$ after illumination for 300 s following treatment with AZ. **d** Mean rate of $O_2$ evolution relative to control following treatment with AZ. Error bars represent s.e.m.

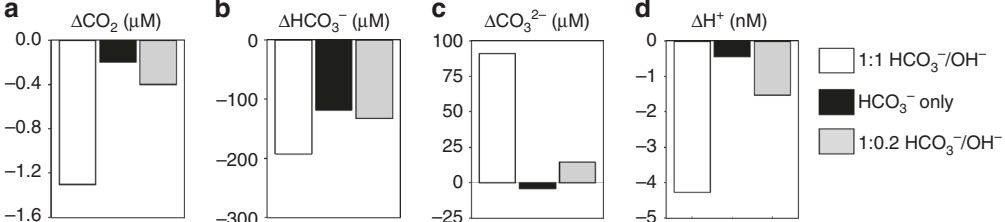

**Fig. 4** Cellular modelling of $HCO_3^-$ uptake. **a–d** Simulated changes in concentrations of inorganic carbon species and $H^+$ at the cell surface relative to bulk solution concentrations. Three different scenarios were simulated: (1) Photosynthesis supported by $HCO_3^-$ uptake with equimolar $OH^-$ export, (2) $HCO_3^-$ uptake only, (3) $HCO_3^-$ uptake with $OH^-$ export occurring at 20% the rate of $HCO_3^-$ uptake, a scenario chosen to best match the $H^+$ drawdown observed in the presence of AZ

MIMS-based approach[15,23]. eCA activity was determined by measuring the removal of $^{18}O$ from $CO_2$, after prior calculation of intracellular CA activity (iCA) and the membrane permeabilities of $CO_2$ and $HCO_3^-$ (Supplementary Fig. 3). We obtained an estimate for eCA activity of $8.3 \pm 1.7 \times 10^{-5}$ cm$^3$ s$^{-1}$ (expressed as the first order rate constant for eCA-catalysed $CO_2$ hydration, $n = 4$), which indicates that *O. sinensis* possesses substantial eCA activity, similar to other large centric diatoms[23].

We next examined how inhibition of eCA affected cell surface pH in *O. sinensis* using the CA inhibitor acetazolamide (AZ). As AZ is only very weakly membrane-permeable, it can be used in short-term studies to specifically inhibit eCAs without affecting the activity of iCAs[38]. 100 μM AZ completely inhibits eCA activity in the centric diatoms *Thalassiosira pseudonana* and *T. weissflogii* without influencing the activity of iCAs[15]. We found that 100 μM AZ had a profound impact on the changes in cell surface pH, inhibiting the light-dependent decrease in $[H^+]$ to $28.6 \pm 7.84\%$ of control cells ($n = 7$ cells; Fig. 3a–c). The inhibitory effect of AZ on cell surface pH changes was rapidly reversed when the inhibitor was removed by perfusion with control ASW media. 100 μM AZ also significantly inhibited the rate of photosynthetic $O_2$ evolution to $57.4 \pm 3.52\%$ of the control ($n = 7$; Fig. 3d). Therefore, in the absence of eCA, the cell is able to maintain photosynthesis at approximately half the initial rate, but the elevations in cell surface pH are greatly decreased. Addition of 10 μM benzolamide (BZA), another member of the sulphonamide class of CA inhibitors, also dramatically decreased the elevations in cell surface pH ($\Delta H^+$ was decreased to $27.6 \pm 2.73\%$ of the control, $n = 4$ cells; Supplementary Fig. 4).

Our results have several important implications for understanding DIC uptake in large phytoplankton cells. First, they support other lines of evidence suggesting that eCA contributes significantly to photosynthetic DIC uptake in large diatoms under standard DIC conditions. Second, they suggest that the activity of eCA is primarily responsible for elevating cell surface pH during photosynthetic DIC uptake. Third, they imply that the mechanism of photosynthetic DIC uptake that supports $O_2$ evolution in the absence of eCA does not substantially influence pH in the cell surface microenvironment. This is most likely to be direct $HCO_3^-$ uptake (although there may also be a small contribution from diffusive $CO_2$ entry) and suggests that $HCO_3^-$ uptake is not closely coupled to $H^+$ uptake (or $OH^-$ efflux). To explore this further, we performed a series of modelling simulations to examine the best fit to our experimental data. We found that an uptake stoichiometry of $0.2H^+$ for each $HCO_3^-$ provided the best fit to the experimental determination of cell surface carbonate chemistry in eCA-inhibited cells (Fig. 4a–d). If $HCO_3^-$ uptake is not directly balanced with equimolar $H^+$ uptake (or $OH^-$ efflux), the source of the additional $H^+$ required for the use of $HCO_3^-$ in carbon fixation is unclear. The shortfall may be met by $H^+$ uptake

that is temporally uncoupled from $HCO_3^-$ uptake or by $H^+$ generation through other metabolic processes.

**Inhibition of eCA at low DIC.** Experiments with small diatoms suggest that eCA-mediated $CO_2$ uptake makes a much larger contribution to photosynthetic DIC uptake under conditions of low DIC[15]. We therefore examined the microenvironment around *O. sinensis* in artificial seawater (ASW) media containing 0.5 or 2 mM DIC. Photosynthetic $O_2$ evolution was slightly lower at 0.5 mM DIC (mean rate of $O_2$ evolution $89.5 \pm 2.6\%$ of control, $n = 3$ cells), but the light-dependent decreases in cell surface $[H^+]$ were greatly enhanced (Fig. 5a–d). This is likely due to the decreased buffering capacity at low DIC, but could also reflect an increased requirement for eCA due to the lower availability of $CO_2$. However, the proportion of $O_2$ evolution that was inhibited by 100 μM AZ was not increased at 0.5 mM DIC ($31.2 \pm 1.36\%$ inhibition, $n = 8$) compared to cells at 2 mM DIC ($43.6 \pm 3.52\%$, $n = 7$). Simulations using the cellular model indicated a decrease in cell surface $[H^+]$ at 0.5 mM DIC of 3.3 nM compared to 1.9 nM at 2 mM DIC, assuming a fixed photosynthetic and DIC uptake rate. These results suggest that the greater increase in cell surface pH observed at low DIC is primarily due to the decrease in buffering capacity. As DIC is the primary contributor to the buffering capacity of seawater, the buffer capacity ($\beta$) is greatly reduced at 0.5 mM DIC.

Detailed examination of the kinetics of $O_2$ evolution revealed a pronounced additional effect of inhibiting eCA at low DIC. In the presence of 100 μM AZ, $O_2$ initially rises rapidly at the onset of illumination in a manner similar to the control, but reaches a transient peak after only $21.7 \pm 0.67$ s, which is followed by a much slower rise to a stable value ($n = 7$; Fig. 5c). The transient peak in cell surface $O_2$ may therefore reflect initial photosynthetic activity supported by diffusive $CO_2$ entry before $CO_2$ becomes rapidly depleted at the cell surface due to the inhibition of eCA.

**Carbonate chemistry in the cell surface microenvironment.** Further understanding of the carbonate chemistry in the microenvironment around a diatom cell requires knowledge of other parameters of the carbonate system in addition to pH. Measurement of $[CO_3^{2-}]$ can give valuable insight into the nature of dynamic changes in carbonate chemistry because the equilibration between $CO_3^{2-}$ and $HCO_3^-$ is very rapid, especially when compared to the uncatalysed equilibration between $HCO_3^-$ and $CO_2$[2]. We therefore used a $CO_3^{2-}$-selective microelectrode in combination with a pH microelectrode to simultaneously measure both of these parameters at the surface of a single *O. sinensis* cell (Supplementary Fig. 5). We found that $[CO_3^{2-}]$ rapidly became elevated at the cell surface upon illumination, with kinetics that closely matched the increase in pH (Fig. 6a, b). The

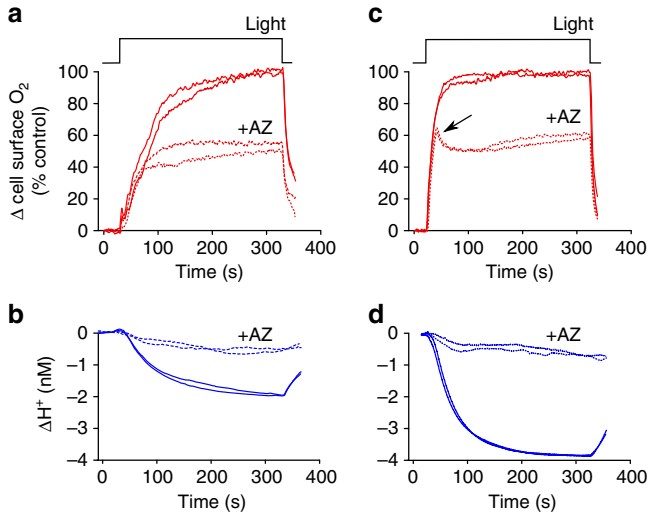

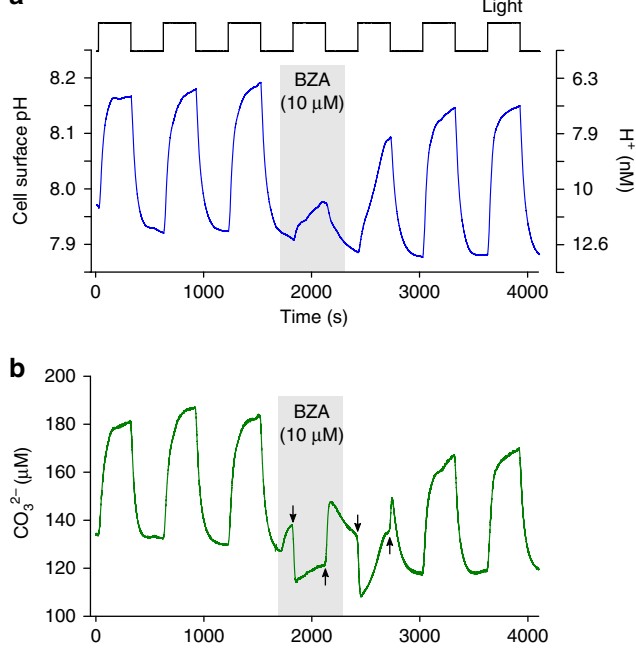

**Fig. 5** The cell surface microenvironment under low DIC conditions. **a** Detailed view of the increase in cell surface $[O_2]$ following illumination of an *O. sinensis* cell in ASW media containing 2 mM DIC at pH 8.0 with or without the addition of 100 μM AZ. The $O_2$ traces are shown as % of the untreated control. **b** The decrease in cell surface $[H^+]$ for cells shown in **a**. **c** Cell surface $[O_2]$ around an *O. sinensis* cell in ASW media containing 0.5 mM DIC at pH 8.0 with or without the addition of 100 μM AZ. In the presence of AZ at 0.5 mM DIC, the initial rise in $[O_2]$ at the cell surface is very similar to the untreated control, but this rate cannot be sustained and falls to a lower level. **d** The decrease in cell surface $[H^+]$ for cells shown in **c**. The depletion of $[H^+]$ at the cell surface is much greater at 0.5 mM DIC than at 2 mM DIC. Representative traces are shown from two individual cells, $n = 7$ cells examined

very close relationship between pH and cell surface $[CO_3^{2-}]$ suggests that the changes in $[CO_3^{2-}]$ are driven directly by the changes in pH, which act to shift the chemical equilibrium towards $CO_3^{2-}$ as predicted by our cellular model of eCA-catalysed $CO_2$ uptake (Fig. 2). The elevations in $[CO_3^{2-}]$ were strongly inhibited in the presence of eCA inhibitors (100 μM AZ or 10 μM BZA) (Figs. 6a, b and 7a–d). The experimental data therefore suggest that sequestration of $H^+$ during the eCA-catalysed conversion of $HCO_3^-$ to $CO_2$ is primarily responsible for perturbations in carbonate chemistry around an *O. sinensis* cell in the light.

Closer inspection of the experimental data revealed unexpected fine-scale dynamics of $[CO_3^{2-}]$ at the cell surface. In the presence of AZ or BZA, only a very small light-dependent increase in cell surface pH was observed. However, $[CO_3^{2-}]$ did not increase and therefore no longer directly mirrored cell surface pH. Instead, there was a pronounced and immediate decrease in $[CO_3^{2-}]$ at 'light on' and a similar rapid increase in $[CO_3^{2-}]$ at 'light off' ($n = 7$ for AZ and $n = 3$ for BZA; Figs. 6a, b and 7a–d). The speed of these changes suggests that they occur as the direct result of a light-activated process. The depletion of $[CO_3^{2-}]$ may therefore result from rapid equilibration between $CO_3^{2-}$ and $HCO_3^-$ at the cell surface following light-dependent activation of $HCO_3^-$ uptake. In support of this, our cellular model demonstrates that $HCO_3^-$ uptake in the absence of significant changes in pH leads to a depletion of $[CO_3^{2-}]$ at the cell surface (Fig. 4c). Direct light-dependent activation of $HCO_3^-$ uptake has previously been observed in cyanobacteria[39].

**Seawater carbonate chemistry influences the microenvironment.** Previous researchers have indicated that the growth of

**Fig. 6** Simultaneous measurement of pH and $CO_3^{2-}$ at the cell surface. **a** Light-dependent changes in cell surface pH around an *O. sinensis* cell. **b** Light-dependent changes in cell surface $[CO_3^{2-}]$ around the cell described in **a**. In the untreated cell, illumination results in a rapid increase in cell surface $[CO_3^{2-}]$ that very closely mirrors the rise in pH. On addition of 10 μM benzolamide (BZA), the increase in cell surface pH is dramatically reduced and $[CO_3^{2-}]$ no longer mirrors pH but shows an immediate decrease upon illumination (down arrow), which is restored after the cell is returned to the dark (up arrow). Following the removal of BZA, the light-dependent increase in cell surface pH is rapidly restored, although the decrease in $[CO_3^{2-}]$ persists for one light/dark cycle (arrowed). Note a small increase in $[CO_3^{2-}]$ coincides with the addition of BZA, which is due to slight differences in the carbonate chemistry of the ASW media containing BZA

large diatoms is significantly enhanced under elevated $CO_2$ conditions, although the cellular mechanisms responsible remain unclear[17]. We therefore investigated how similar changes in seawater carbonate chemistry may influence the microenvironment around single *O. sinensis* cells, measuring cell surface pH and $[CO_3^{2-}]$ in ASW adjusted to pH 7.6, pH 8.2 or pH 8.8 by $CO_2$ bubbling. We found that the light-driven change in cell surface $[H^+]$ is much greater at pH 7.6 than at pH 8.2 (Fig. 8a–c), a key prediction of previous modelling studies[36]. In contrast, the cell experiences a much narrower range of cell surface $[CO_3^{2-}]$ at pH 7.6 compared to pH 8.2. At pH 8.8, the range of both $[H^+]$ and $[CO_3^{2-}]$ experienced at the cell surface is lower than the range experienced at pH 8.2. Although bubbling with $CO_2$ does not change the total alkalinity of seawater, the buffer capacity ($\beta$) is lowered as the pH shifts away from optimal buffer capacity provided by the $HCO_3^-/CO_3^{2-}$ equilibrium (pK$_2$ of carbonic acid in seawater is approximately 9 at 20 °C)[2].

The experimental observations were closely mirrored by simulations of these experiments using our cellular model (which assumed fixed rates of photosynthesis and DIC uptake, Supplementary Fig. 6). Simulations at high $CO_2$ (35 μM, pH 7.6) indicate that eCA is still required to minimise depletion of $[CO_2]$ at the cell surface, even though $CO_2$ availability is much greater (Supplementary Fig. 6). Growth at elevated $CO_2$ is therefore unlikely to circumvent the requirement for eCA in large

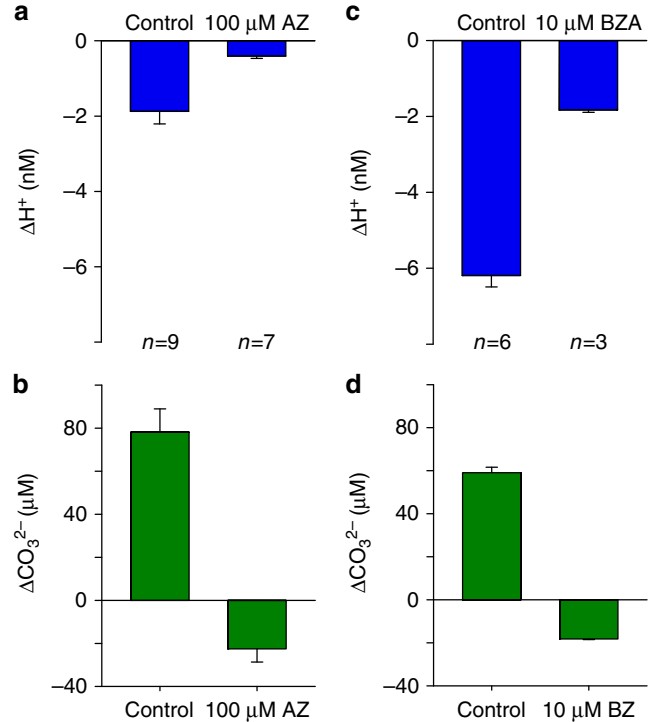

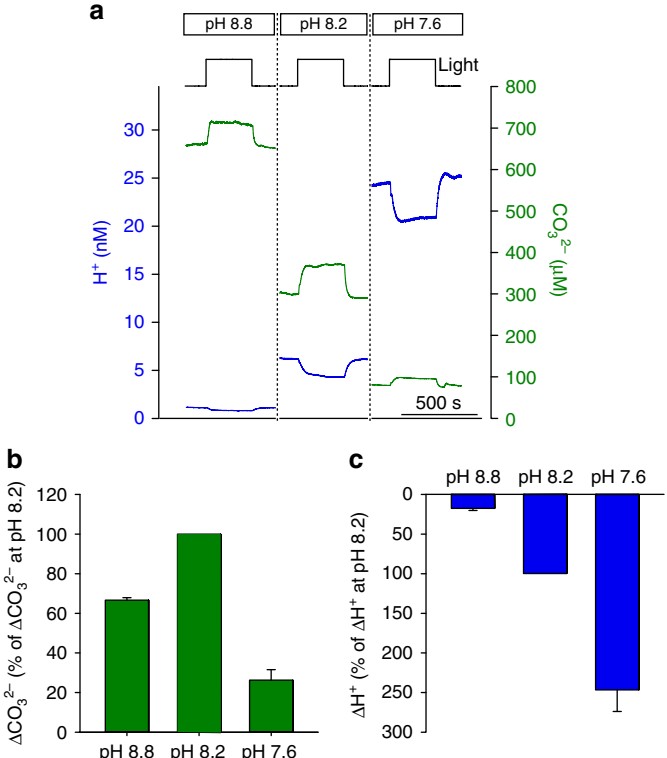

**Fig. 7** The effect of eCA inhibitors on cell surface pH and $CO_3^{2-}$. **a** Mean changes in cell surface $[H^+]$ after illumination for 300 s at pH 8.2 following treatment with 100 μM AZ. **b** The mean changes in $[CO_3^{2-}]$ for cells described in **a**. **c** Mean changes in cell surface $[H^+]$ after illumination for 300 s at pH 8.0 following treatment with 10 μM BZA. **d** The mean changes in $[CO_3^{2-}]$ for cells described in **c**. Error bars represent s.e.m.

diatoms. The model also indicates that a lower $\Delta[CO_3^{2-}]$ is expected at pH 8.8, compared to pH 8.2. This result suggests the observed changes in cell surface carbonate chemistry are primarily due to differences in seawater carbonate chemistry and that the lower $\Delta[CO_3^{2-}]$ at pH 8.8 is a result of the very small change in cell surface pH that occurs under these conditions. Together, the results illustrate that the nature of the microenvironment around diatom cells is highly dependent on the carbonate chemistry of the surrounding bulk seawater.

## Discussion

The transport processes underlying photosynthetic DIC uptake are complex as $CO_2$ represents such a low proportion of the available DIC and the uncatalysed interconversion of $HCO_3^-$ and $CO_2$ is slow and highly pH-dependent. This complexity has led to uncertainty over the mechanisms of photosynthetic DIC uptake in diatoms and in particular the role of eCA in enhancing the supply of $CO_2$ to the cell surface[40]. Our direct measurements of *O. sinensis* indicate that carbonate chemistry in the microenvironment around this large diatom is strongly influenced by photosynthetic DIC uptake and that eCA plays a major role in this process. The significant increases in cell surface pH and $[CO_3^{2-}]$ in the light and the sensitivity of these processes to eCA inhibitors are consistent with a role for eCA in enhancing the supply of $CO_2$ to the cell surface.

Our data do not support the proposed alternative roles for eCA in minimising diffusive loss of $CO_2$ or regulation of cell surface pH[16,24]. Substantial diffusive loss of $CO_2$ can be detected in organisms that lack eCA and accumulate DIC through active $HCO_3^-$ uptake, such as the cyanobacterium *Synechococcus*[41]. If eCA acted to minimise diffusive $CO_2$ loss during active $HCO_3^-$ uptake in diatoms, then its activity would lower cell surface pH in

**Fig. 8** Cell surface carbonate chemistry at altered seawater pH. **a** Simultaneous measurement of cell surface $[H^+]$ and $[CO_3^{2-}]$ around an *O. sinensis* cell in ASW media at pH 8.8. Measurements around the same cell were then taken after the ASW media was bubbled with $CO_2$ to reduce the pH sequentially to 8.2 and 7.6. **b** Mean light-dependent changes in cell surface $[CO_3^{2-}]$ in ASW media at pH 8.8, 8.2 and 7.6. **c** Mean light-dependent changes in $[H^+]$ for the cells shown in **b**. The results are shown as percentage of the light-dependent changes observed at pH 8.2 to normalise for variability in the photosynthetic activity between individual cells. $n = 3$ cells. Error bars represent s.e.m.

the light (although the effect is small, Fig. 2h), whereas we observe a substantial increase. Similarly, there is no evidence to suggest that eCA plays a role in maintaining cell surface pH, as eCA activity is a major contributor to the changes in cell surface pH in the illuminated cells. We therefore conclude that the primary role for eCA in *O. sinensis* is to enhance the supply of $CO_2$ to the cell surface by catalysing the conversion of $HCO_3^-$ to $CO_2$. Our data suggest that this process contributes significantly to overall photosynthetic DIC uptake and is likely to play a critical role in overcoming spatial and temporal variability in the supply and demand of $CO_2$.

As eCA acts to catalyse the equilibration between $HCO_3^-$ and $CO_2$, its activity alone cannot increase $[CO_2]$ inside the cell above that of the bulk seawater. Other processes, such as active transport of $HCO_3^-$ into the chloroplast (the chloroplast pump model)[42] or the activity of C4 biochemical CCM[14], allow the cell to accumulate $CO_2$ at the site of RuBisCO and also serve to keep DIC concentrations low in the cytosol. This helps to maintain an inward diffusive influx of $CO_2$ across the plasma membrane. In the absence of eCA, diffusive limitation of $CO_2$ supply to the surface of *O. sinensis* would require a prohibitively low $[CO_2]$ in the cytosol in order to generate an inward diffusive $CO_2$ gradient. As the cell is not spherical, there is also spatial variability in the diffusive supply of $CO_2$ to the cell surface, which would result in large differences in the $CO_2$ gradient across the plasma membrane

in the absence of eCA. Temporal variability in $CO_2$ supply and demand through rapid fluctuations in irradiance caused by turbulent mixing, wave focussing and/or turbidity would also lead to problems in maintaining an inward $CO_2$ gradient across the plasma membrane in the absence of eCA[43]. The direct light-dependent activation of eCA activity observed in the diatom *Skeletonema* suggests that eCA contributes to a rapid response of the CCM to changes in irradiance[44]. The expression of eCA therefore allows large, irregular-shaped cells to maintain $[CO_2]$ at the cell surface during fluctuations in the supply and demand for $CO_2$ and overcome the potential limitation caused by their diffusive boundary layer.

Given the broad distribution of eCA among marine diatoms and other phytoplankton[18], it is likely that eCA performs a similar role in many other species, especially in larger cells. In particular, eCA activity appears to be ubiquitous in centric diatoms, suggesting that it plays a conserved role in DIC uptake in this lineage[23]. However, clear trends supporting the requirement for eCA have not emerged from previous experimental analyses[9,27]. The morphology, physiology and ecology of each species may all contribute to variability in the requirement for eCA, for example by influencing growth rate and the subsequent demand for DIC uptake to sustain carbon fixation. Most diatom species that exhibit a requirement for eCA at ambient DIC are sufficiently large to likely encounter diffusive limitation of $CO_2$[23,45]. Although eCA activity may have some energetic benefits for smaller diatoms at typical oceanic $CO_2$ concentrations by minimising $CO_2$ depletion at the cell surface[23,46], these species only demonstrate a clear requirement for eCA when DIC becomes limiting[15].

The relative proportions of $CO_2$ and $HCO_3^-$ uptake across the plasma membrane are likely to be strongly influenced by both the supply and the demand for DIC[9]. We found that inhibition of eCA in *O. sinensis* decreased photosynthetic $O_2$ evolution to approximately half the control rate, indicating that other transport processes (primarily active $HCO_3^-$ transport) act to supply DIC at a similar rate to eCA-catalysed $CO_2$ diffusion. Previous estimates of the contribution of eCA to DIC uptake have generally assumed that eCA inhibition does not lead to rapid compensatory changes in other DIC uptake processes. For example, MIMS measurements of DIC uptake in phytoplankton are routinely made in the presence of eCA inhibitors to allow discrimination between $CO_2$ and $HCO_3^-$ uptake. However, our microelectrode measurements provide some evidence to suggest that $HCO_3^-$ uptake may be stimulated by the absence of eCA. Certainly, we observed a light-dependent depletion of $[CO_3^{2-}]$ in the absence of eCA that is most likely caused by rapid activation of $HCO_3^-$ uptake. As this was not observed when eCA was active, this proportion of $HCO_3^-$ uptake may be specifically activated by DIC limitation following the inhibition of eCA. Thus, a significant increase in $HCO_3^-$ uptake may occur in the presence of eCA inhibitors, which could result in underestimation of the contribution of eCA-catalysed $CO_2$ supply to DIC uptake in many analyses.

Previous analyses using MIMS and the isotope disequilibrium technique led to suggestions that eCA does not function primarily in maintaining $CO_2$ supply, because diatoms with high eCA activity show a much greater proportion of $HCO_3^-$ uptake relative to $CO_2$ uptake[16,21]. However, these findings should be interpreted cautiously. The application of MIMS to compare the relative proportion of $HCO_3^-$ and $CO_2$ uptake requires the presence of an eCA inhibitor and therefore does not measure the proportion of $CO_2$ uptake catalysed by eCA[27,47]. In a large diatom with a significant diffusive boundary layer, ignoring eCA-catalysed $CO_2$ supply could lead to a considerable overestimation of the contribution of direct $HCO_3^-$ uptake, especially if eCA

inhibition also leads to an activation of $HCO_3^-$ uptake as noted above. Whilst isotopic disequilibrium approaches can potentially discriminate between eCA-catalysed conversion of $HCO_3^-$ and direct $HCO_3^-$ uptake in the absence of an eCA inhibitor[48], these two processes are difficult to distinguish if eCA activity is high, which may result in underestimation of the contribution of eCA.

The sulphonamide class of carbonic anhydrase inhibitors have been used extensively to inhibit eCA as they exhibit only very weak membrane permeability. Our direct observations of rapid inhibition of eCA in single *O. sinensis* cells by AZ or BZ followed by a rapid recovery are consistent with no internalisation of these inhibitors. A direct inhibitory effect of AZ on plasma membrane $HCO_3^-$ transporters was proposed following application of the isotope disequilibrium technique to marine diatoms[21], although the evidence for this was largely indirect. Subsequent experimental analyses have demonstrated that 100 μM AZ has no impact on the activity of the SLC4-2 $HCO_3^-$ transporter of *Phaeodactylum tricornutum*[11]. Moreover, the evidence from our microelectrode work supports activation rather than inhibition of $HCO_3^-$ transport in the presence of eCA inhibitors.

The development of $CO_3^{2-}$ ionophores that exhibit sufficient selectivity for $CO_3^{2-}$ over the other major anions in seawater (particularly $Cl^-$) has enabled the development of microsensors that can robustly and reliably measure $[CO_3^{2-}]$ in seawater[49,50]. $CO_3^{2-}$ microelectrodes have been used previously to demonstrate that $[CO_3^{2-}]$ is elevated around the very large cells of foraminifera (*Amphistegina* sp.) during photosynthesis[51] and more recently to demonstrate that $[CO_3^{2-}]$ is greatly elevated in the calcifying fluid in the internal cavity of a coral polyp[52]. These results, together with our own, demonstrate that the ability to measure pH and $[CO_3^{2-}]$ can provide important information on spatiotemporal variability in carbonate chemistry around marine organisms and its impact on their physiology. If the carbonate system is at equilibrium, measurement of two parameters is commonly used to calculate the other parameters. However, our model of cell-surface DIC chemistry demonstrates that the carbonate system at the surface of a phytoplankton cell is dominated by fluxes across the plasma membrane and is not at equilibrium. In this case, parameters that are not measured cannot be solely derived from equilibrium constants, which is an important consideration for future research into the influence of the microenvironment on cell physiology[31].

Much recent research interest has focused on the impact of predicted future changes in ocean pH on phytoplankton physiology[47]. Seawater pH is very stable in open ocean environments and phytoplankton inhabiting these environments do not experience significant changes in bulk seawater pH[53]. However, it is clear that larger diatoms are likely to experience significant variability in cell surface pH even if they inhabit areas where bulk seawater pH is stable. As this variability is primarily a consequence of eCA activity, maintaining a stable $[CO_2]$ at the cell surface appears to be of greater benefit than maintaining a stable pH, at least in the short term. The carbonate chemistry experienced by a diatom cell will be highly dependent on irradiance, cell size/morphology and the chemistry of the bulk seawater. Changes in cell physiology, such as the requirement for eCA in a high $CO_2$ environment, will also provide a major influence on the carbonate chemistry experienced by a cell. Previous experimental evidence indicates that there is a general trend for the proportion of DIC taken up as $HCO_3^-$ to decrease with increased $CO_2$ availability[9,28]. Our simulations show that eCA is still required by large diatoms at elevated $CO_2$, although the level of eCA activity required will be lower. This could contribute to savings in the cellular energy budget and contribute to the specific growth enhancement of large diatoms at elevated $CO_2$[17], but estimations suggest that eCA represents only a very small proportion of the

cellular nitrogen budget in diatoms[23]. It will therefore be important to examine whether the growth enhancement in large diatoms at elevated $CO_2$ is dependent on the requirement for eCA or derives from other potential savings to cellular energy budgets (e.g., decreased diffusive loss of $CO_2$ or a switch from active $HCO_3^-$ uptake to passive $CO_2$ uptake).

It is becoming increasingly clear that in order to better understand phytoplankton physiology we cannot just consider nutrient transport processes on a bulk scale, but must examine the chemical, physical and biological processes that occur within the microenvironment around the cell or 'phycosphere'[54]. Our results demonstrate that the microenvironment around a single diatom cell is extremely dynamic. We show that pH can change dramatically at the cell surface within seconds, that pH varies across the surface of the cell and that the microenvironment extends considerably away from the cell. Whilst phytoplankton cells typically remain smaller than the smallest turbulent eddies, the scale of the boundary layer is sufficiently large to be disrupted by these eddies[55], suggesting that large cells could benefit from increased $CO_2$ supply in areas of strong turbulence. Movement of large cells through the water column due to sinking may also act to reduce the size of the boundary layer and increase the diffusive supply of $CO_2$ to the cell surface. Our measurements provide clear evidence for a role for eCA in the maintenance of [$CO_2$] at the cell surface, in order to enhance the supply of $CO_2$ for photosynthesis, but they also point to significant greater complexity in the regulation of DIC uptake. Future elucidation of these regulatory mechanisms will greatly further our understanding of the process of carbon assimilation in diatoms and in other marine phytoplankton.

## Methods

**Algal strains and culturing conditions**. *O. sinensis* (strain PLY624), *O. mobiliensis* (PLY618) and *Thalassiosira weissflogii* (PLY541) were obtained from Plymouth Culture Collection of Marine Microalgae. *Coscinodiscus* sp was isolated from seawater samples collected from station L4, Western English Channel in November 2014. Cultures were maintained in aged filtered seawater with f/2 media with 100 μM silicate[56] under irradiance of 80–100 μmol s$^{-1}$ m$^{-2}$, with a temperature of 15 °C and a photoperiod of 18:6 h light:dark. *O. sinensis* cells were maintained at a low cell density (<50 cells per mL) and the pH of the culture medium was routinely measured to ensure that the cells did not experience significant changes in carbonate chemistry in their culture vessels (culture pH was maintained between 8.1 and 8.3). All experimental analyses were performed in ASW prepared as described previously[57]. Unless otherwise mentioned, ASW solutions were not buffered and pH$_{NBS}$ was adjusted by addition of HCl or NaOH. The total alkalinity was measured by Gran titration and was typically 2500–2600 μmol kg$^{-1}$. All chemicals were obtained from Sigma, unless otherwise stated. Benzolamide was a gift from Dr Juha Voipoi (University of Helsinki), from an original stock synthesised by Dr E.R. Swenson (University of Washington, Seattle, WA, USA). The addition of certain inhibitors, particularly at millimolar concentrations, can significantly change the carbonate chemistry of unbuffered seawater, which was a major consideration in our choice of inhibitors and their effective concentration.

**Microelectrode fabrication and calibration**. The ion-selective microelectrodes were prepared in a similar manner to those described previously[50,51]. Briefly, borosilicate glass capillaries (length 150 mm, outer diameter 1.5 mm, inner diameter 1.17 mm) were pulled to a fine point using a P-97 pipette puller (Sutter, Novato, CA, USA). For large cells, blunt-end electrodes were prepared by fire polishing with an outer diameter of ~20 μm and an inner diameter of 1–2 μm (Supplementary Fig. 5). The capillaries were then silanised by exposure to *N*,*N*-dimethyltrimethylsilylamine vapour at 200 °C for 1 h. The pH microelectrodes were prepared by filling with hydrogen ionophore I—cocktail A (Sigma) containing hydrogen ionophore I (10.0 % wt), 2-nitrophenyl octyl ether (89.3 % wt) and sodium tetraphenylborate (0.7 % wt). The filling solution was 100 mM NaCl, 20 mM HEPES pH 7.2 and 10 mM NaOH. The reference electrode was filled with 3 M KCl. Data were recorded using an AxoClamp 2B amplifier, with pClamp v9 software (Molecular Devices, CA, USA). Each pH electrode was calibrated using buffered artificial seawater standards (10 mM HEPES) adjusted to pH 7.0, 8.0 and 9.0 by the addition of HCl or NaOH. The pH$_{NBS}$ in each seawater standard was determined using a Ross Orion electrode. The slope of the calibrated electrodes ranged from 51–57 mV/pH unit (Supplementary Fig. 7).

The $CO_3^{2-}$ microelectrodes were prepared using an ionophore cocktail containing *N*,*N*-dioctyl-3α,12α-bis(4-trifluoroacetylbenzoyloxy)-5β-cholan-24-

amide (11% wt), tridodecylmethylammonium chloride (4% wt), 2-nitrophenyl octyl ether (75% wt) and polyvinyl chloride (10%) as described by Han et al.[50]. The filling solution was 19.1 g L$^{-1}$ Na$_2$B$_4$O$_7$.10H$_2$O as described by de Beer et al.[51]. The reference electrode was a glass capillary containing a Ag/AgCl wire and filled with 3 M KCl. The $CO_3^{2-}$ electrodes were calibrated using a three-point calibration. ASW solutions at three different pH values (pH 8.8, 8.2 and 7.6) were prepared by adjusting pH by bubbling with $CO_2$ until the pH had stabilised and fully equilibrated. The pH$_{NBS}$ and total alkalinity were determined for each standard using a Ross Orion electrode and Gran titration, respectively. The $CO_3^{2-}$ concentration in each standard was then calculated using the CO2SYS program with constants from Roy et al.[58] and ranged from 80–700 μmol kg$^{-1}$. The response of the $CO_3^{2-}$ microelectrodes was log-linear to [$CO_3^{2-}$] within this range and the calculated slopes ranged from 28 to 31 mV per decade, which is similar to previous $CO_3^{2-}$ microelectrodes fabricated in this manner[50,51] (Supplementary Fig. 8).

$O_2$ measurements were performed using a Firesting $O_2$ optode with 50 μm tip diameter (Pyroscience, Aachen, Germany). It should be noted that the $O_2$ microsensor was significantly larger than the ion-selective microelectrodes and therefore measured $O_2$ at a different spatial resolution to the ion-selective microelectrodes. The $O_2$ measurements were used to measure relative changes in photosynthetic rate and were not incorporated into the cellular models, so the spatial resolution of the $O_2$ sensor was not required to be of the same scale as the other microelectrodes. The $O_2$ optode was calibrated according to manufacturer's instructions. For simplicity, a simple two-point calibration using 0 and 100 % air saturated solutions (using 30 g L$^{-1}$ sodium dithionite to produce a 0% $O_2$ solution) was used for most samples. These $O_2$ data are therefore presented as percentage air saturation rather than quantitative measurements of $O_2$ and were used to examine the effect of a treatment on an individual cell, rather than to compare the effects of treatments between groups of individuals. The relative photosynthetic rates were estimated from $O_2$ measurements using the method of Revsbech et al.[59], which assumes that the rate of $O_2$ evolution in the light is equivalent to the rate of the decrease in $O_2$ concentration at the very start of the dark period.

**Microelectrode measurements of the cell surface**. Cells were placed on a glass-bottomed microscopy dish (35 mm diameter) and observed using an Axiovert A.1 inverted microscope (Zeiss). Unless otherwise stated, cells were illuminated at 200 μmol m$^{-2}$ s$^{-1}$ using an OptoLED lite white light source (Cairn). Temperature was monitored throughout and was maintained at 20 ± 1 °C. There was no change in temperature within the dish during illumination. Microelectrodes were positioned directly against the cell using a MP-225 micromanipulator (Sutter). In experiments where two microelectrodes were used simultaneously, two micromanipulators were used to position each microelectrode on opposing sides of the cell. To standardise our measurements, we positioned the pH microelectrodes at the centre of the cell unless otherwise stated. However, it should be noted that that comparisons of boundary layer dynamics between different cells and different species must carefully consider the influence of morphology and the positioning of the microelectrodes. The cells were perfused with ASW at a constant flow rate of 1 mL min$^{-1}$ throughout and treatments were added by perfusion. The volume of ASW in the recording dish was 2.5 mL.

Experimental analyses were all performed using cells placed directly on the glass bottom dish. To test whether the dish significantly influenced the formation of the diffusion boundary layer around cells, we also measured cell surface pH in cells that were suspended on a fine mesh (pore size 100 μm) to allow diffusion in all directions. The results were very similar to those observed when cells placed on the dish, with cells exhibiting rapid increases in cell surface pH upon illumination (Supplementary Fig. 9), suggesting that the dish did not have a major influence on the formation of the diffusive boundary layer or the underlying physiological processes.

**Modelling the cell surface microenvironment**. A model describing cell surface DIC chemistry was developed based on the spherical reaction-diffusion model used by Hopkinson et al.[15]. Briefly, this model enables determination of carbonate chemistry at the cell surface, through the diffusive boundary layer (100–300 μm thick), and into the bulk seawater. The model assumes a constant concentration of each component in the bulk solution but allows these to vary in proximity to the cell surface due to the various uptake and export fluxes and reaction-diffusion within the boundary layer. eCA is assumed to act only at the cell surface. The model includes dissolved DIC species ($CO_2$, $HCO_3^-$, and $CO_3^{2-}$) and other important components for determining seawater pH such as, $H^+$, $OH^-$, $B(OH)_3$, and $B(OH)_4^-$, using rate constants from Zeebe and Wolf-Gladrow[2]. The following parameters were used for the model unless otherwise stated: photosynthetic rate $8 \times 10^{-14}$ mol C per cell per s, eCA activity (when active) $8.3 \times 10^{-5}$ cm$^3$ s$^{-1}$, pH 8.0, DIC 2 mM, temperature 20 °C, cell radius 60 μm, salinity 35 g kg$^{-1}$. The value for eCA activity was derived from MIMS measurements (see below), whereas the photosynthetic rate was estimated from the observed pH changes in the absence of an eCA inhibitor. The estimated photosynthetic rate was comparable to measured values in large centric diatoms[23]. The model was solved in Matlab.

**Membrane-inlet mass spectrometry**. The iCA and eCA activities of *O. sinensis* cells were determined using membrane-inlet mass spectrometry (MIMS). Cells were added to assay buffer (C$_i$-free artificial seawater, 20 mM Tris at pH 8.0)

containing $^{18}$O-labelled $^{13}$C-Ci (2 mM, unless otherwise noted) within a MIMS chamber maintained at 20 °C. The rate of $^{18}$O removal from labelled $C_i$ was monitored by MIMS. The results were fitted to a model to determine eCA activity as described previously[15]. Prior calculation of intracellular CA activity (iCA) and the membrane permeabilities of $CO_2$ and $HCO_3^-$ was performed by analysis of $^{18}$O-removal from $CO_2$ in the presence of 50 μM of the eCA inhibitor, dextran-bound AZ (DBAZ) (Ramidus, Sweden). iCA activity ($k_{cf}$) was estimated to be 616 ± 23 s$^{-1}$ and the $CO_2$ and $HCO_3^-$ mass transfer coefficients were $1.60 \times 10^{-6} \pm 0.08 \times 10^{-6}$ cm$^3$ s$^{-1}$ and $3.2 \times 10^{-9} \pm 3.1 \times 10^{-9}$ cm$^3$ s$^{-1}$, respectively. eCA activity was then determined from $^{18}$O-removal from $CO_2$ catalysed by *O. sinensis* cells without an eCA inhibitor.

**Data availability**. The relevant data from this study are available from the authors.

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

## Acknowledgements

We thank Juha Voipio for materials and advice relating to microelectrode fabrication. We also thank Dorothee Kottmeier for discussions and comments on the manuscript. G.L.W., C.B. and K.F. acknowledge funding from the Natural Environment Research Council (NE/J021954/1). B.M.H. acknowledges funding from the US National Science Foundation (EF 1041023 and MCB 1129326).

## Author contributions

A.C., G.L.W. and C.B. designed the experimental work. A.C. performed the microelectrode analyses. A.C., B.M.H. and G.L.W. analysed the experimental data. B.M.H. performed the cellular modelling and MIMS analyses. A.C., B.M.H., K.F., C.B. and G.L.W. wrote the manuscript.

## Additional information

**Competing interests:** The authors declare no competing financial interests.

