## [Peer Review File · Nature Communications]

Reviewers' comments:

Reviewer #1 (Remarks to the Author):

This manuscript addresses an important topic: the role external carbonic anhydrase in the carbon uptake process in marine diatoms. A strength is the marriage of cutting-edge measurement techniques and a sophisticated model. It is disappointing that the manuscript is marred by numerous typos and errors (some of which I have flagged below). The results are detailed and the manuscript would be more effective if the storyline was strengthened to a crisp message.

L43. The issue is not the 1% of CO₂ as DIC but the low (air-equilibrium) concentration relative to Rubisco kinetics. However, the 99% of remaining DIC does represent an opportunity.

L55 – 59 There seem to be some missing statements in the logic here. It would help to be more specific and clarify the link between diffusion limitation and low rate of interconversion of HCO₃⁻ and CO₂.

L66 This is presumably referring to growth rate.

L68 Typo.

L87 I am not sure this was one of the conclusions in this paper.

L96 Typo.

L102 What discrepancies are being referred to? Different species may operate differently anyway. There is also a non-sequitur here: different experimental approaches and underestimation of eCA?

L113-114 Only one of the quoted references appears to relate to foraminifera. How photosynthetic are foraminifera? I understand some have endosymbiotic algae. Some references do not refer to single large cells as implied.

L124 Missing word.

L127 It would be helpful to be more specific than 'to increase the supply of CO₂'

L137 'Significant' in a statistical sense? If not use a different adjective. Check other occasions where 'significant' is used.

L139 Decrease and then a -ve value? Would be easier for most readers if [H⁺] was expressed in pH units (or at least a conversion given).

Figure legends- These are quite wordy and descriptive and do not seem to match the journal style.

L157 Give pH that the cells were buffered at.

L173-174 I would have thought the simplest explanation for the pH increase would have been removal of inorganic carbon in photosynthesis and a consequent increase in pH via the buffering in

the carbonate system. This might be what is meant by conversion of bicarbonate to CO₂ but it could be more clearly expressed.

L182 Does the model assume that eCA is present or effective at distance up to 250 μm from the cell surface?

Fig. 2 What does a negative CO₂ concentration mean? I presume it is linked to the assumption of a fixed uptake rate, but good to clarify.

L192 I have struggled with this and Fig. 2. My first thought was that in the CO₂ uptake simulation, the absence of eCA would generate a higher pH at the cell surface because an out-of-equilibrium situation would have lower CO₂ and higher HCO₃⁻ concentrations than at the new equilibrium. I realise that the model has been described before but it would help clarity and reader comprehension if an additional sentence or two were added in explanation.

L202- Isn't the distinction between diffusion limitation via CO₂ and HCO₃⁻ uptake not to do with concentration but concentration difference? Even though the diffusion coefficient for HCO₃⁻ is about half that of CO₂ in water, the concentration difference generated by an increase in pH is about 20 to 25-times greater for HCO₃⁻ vs CO₂ in your system.

L225 Typo.

General- Both Fig and Fig. are used. Tidy up.

L322 Isn't this smaller range simply down to carbonate chemistry?

General One important point I think not discussed is that although eCA will increase the rate of CO₂ supply it will not elevate the CO₂ concentration at the cell surface above the bulk-water concentration and is therefore not a CCM and so does not solve the problem of low CO₂ compared to Rubisco kinetics given K_m for diatom Rubiscos of 23 to 68 μM reported in Young et al. (2016) JXB doi:10.1093/jxb/erw163.

L361 is wave focussing likely to act at a relevant timescale?

L384 Just under or just over? (57%).

L575 Temperature monitored but it has not been stated explicitly what the temperature was (presumably the 15°C used for growth, but the model and MIMS at 20°C).

The references need tidying up.

Reviewer #2 (Remarks to the Author):

The paper describes experiments with O₂, pH and CO₃²⁻ microsensors on diatoms. Especially the effects of CA inhibitors was tested, for external CA.

Inhibition of CA resulted in reduction of photosynthesis, pH dynamics and CO₃²⁻ dynamics. The inhibitor acts directly or indirectly on photosynthesis, and reduces CO₂ fixation. How this functions is the subject of this paper.

I have not the feeling it is truly understood, that we have progressed our insights in the question why CA activity is needed for DIC uptake. It is a complex issue, where it is easy to get confused. I became confused and still do not know how exactly CA is involved in DIC uptake.

The aim of the measurements should be explicitly formulated. A hypothesis and its test is missing. Also, it may be better explained what the purpose of carbonate sensors is.

It is really interesting that on single cells the surface dynamics are as large as they are recorded here. The work is technically extremely challenging and seems very well done. Some interesting observations are made, such as the O₂ peak in Fig. 4, but we need insight in how reproducible the results are. It must be mentioned how many measurements are made.

DIC is taken up as CO₂ and bicarbonate. The CO₂ supply is rapidly exhausted as the pool is small, thus CA has a role in converting bicarbonate into CO₂. CA inhibitors will thus reduce CO₂ uptake, which could be compensated by bicarbonate uptake. Clearly this compensation does not happen, as bicarbonate is in equilibrium with carbonate, and carbonate dynamics are reduced by CA inhibition. Thus CA is also essential in bicarbonate uptake, or the inhibitor inhibits the bicarbonate uptake. The question is now: how does AZ inhibit photosynthesis? Via DIC uptake most likely, but exactly how? Why is CA, thus the hydration of CO₂ to bicarbonate, important for DIC uptake?

Could AZ have a direct effect on the bicarbonate uptake? That would be the easiest explanation, but it does not provide insight in the processes. The authors have not been able to find the role of CA in DIC uptake.

It seems to me that the data are a bit over-interpreted. I do not follow the note that pH and CO₃²⁻ dynamics are driven by the activity of CA. They are driven by photosynthesis. CA only brings the carbonate system, incl. pH, closer to equilibrium.

In some occasions the authors used H⁺ concentrations, in others pH, so $-\log[H^+]$. Essentially the same but written differently. Stick to one notation, preferably pH.

On several occasions an increase in CO₂ was reported at the surface of phototrophs upon CA inhibition. This could be part of the discussion.

The discussion is rather unstructured and too long. The discussion should better focus on the role of CA in DIC uptake and explain why AZ reduces DIC uptake and photosynthesis. The thinking about irregular shapes and fluctuations is not essential for the concept. The methodological section and the carbonate environment are superfluous.

L132 the MBL does not directly influence processes, but processes together with the MBL change concentrations at the cell surface.

L207 This seems a crucial section. The CO₂ is depleted due to the pH shift? eCA activity during photosynthesis leads to CO₂ depletion due to the pH shift and ensuing shift in carbonate equilibrium towards bicarbonate. It may indeed also be that the bicarbonate depletion by uptake leads to CO₂ conversion to bicarbonate and a pH increase. Inhibiting CA will then not lead to less DIC uptake (as bicarbonate is not limiting) and hence to less photosynthesis.

L244 the pH dynamics are lower, as P is lower. How DIC is taken up can indeed not change the pH at the surface as the pH is in the end controlled by the net CO₂ fixation rate, regardless of the mechanism of DIC uptake.

L263-271 A bit too detailed info. How often have these experiments been repeated? How significant are the data?

The peak is due to a rapidly disappearing stored pool of DIC. That pool should also be there at 2 mM DIC. Why does the O₂ level increase again after $t=100$ s?

It is better to provide the absolute O₂ concentrations instead of % of the untreated control. What is the untreated control? Clearly 100% is not the sample without AZ (the solid line), as that varies as well. So what is 100%?

L296 This should be mentioned earlier: carbonate dynamics provide direct info in the bicarbonate uptake. An advantage over CO₂ microsensors, as CO₂ is not closely coupled to the DIC system.

L341-345 contradictory. If bicarbonate is taken up the equilibrium will shift from CO₂ to bicarbonate leading to pH increase. How can CA prevent CO₂ loss? Loss where to and from where? This concept must be better introduced, discussed and explained.

If CA is inhibited CO₂ increases outside phototrophs.

L344 Please describe the process that leads to the pH excursion. CO₂ fixation leads to the pH increase. CA seems rather responsible for the DIC supply, as inhibiting it reduces the photosynthesis strongly. pH increase alone does not normally do that.

L366 means that CO₂ in the cells is lower than outside. This is unlikely and would seriously make Rubisco ineffective. Kaplan and others have shown that CO₂ is higher inside. What is the flexible response by CA to fluctuations?

Reviewer #3 (Remarks to the Author):

General comments:

This manuscript by Chrachri et al. describes carbonate chemistry at the surface of single diatom cells based on microsensor measurements of pH, CO₃²⁻ and O₂. The results are further analyzed with a model of cellular carbon fluxes to infer information on the mode of carbon uptake, concluding that eCA plays an important role in carbon uptake in *O. sinensis* by accelerating CO₂ supply in the boundary layer.

Overall, it is a well written, convincing manuscript that provides important new insights going beyond a mere description of boundary layer carbonate chemistry. By combining these measurements with very instructive inhibitor experiments and model calculations, the authors are able to significantly advance our understanding of carbon uptake mechanisms in diatoms. However, I have one major question/concern regarding the methodology which needs to be clarified, as well as several more specific comments and questions:

- Regarding the methodology, I wonder how placing the cells on the solid bottom of a Petri dish affects concentrations of O₂ and H⁺ in the boundary layer as compared to a free-floating cell. I am concerned that by preventing diffusion to/from below the cell, the glass bottom may distort concentrations at the cell surface, gradients in the boundary layer (Fig. 1D) and photosynthesis estimates by the Revsbech et al. approach, as demonstrated for phytoplankton aggregates (Ploug and Jorgensen 1999 MEPS). This depends on diffusion within/through the cell which is difficult to estimate, but the magnitude of this effect could be tested e.g. by comparing gradients between cells placed on agar vs the solid glass surface (as the diffusion coefficient for O₂ is the same in 1% agar as in water, see e.g. Ploug et al. 2010, ISME J).

- I find it surprising that under dark conditions, pH and O₂ at the cell surface were virtually the same as in the bulk (l. 134-136). Furthermore, it is interesting that O₂ concentrations decrease below air saturation in the dark when an eCA inhibitor is present (Fig. 3A). How could this be explained?

- L. 245 HCO₃⁻ uptake in *O. sinensis* is suggested here to consist of two different uptake modes (one H⁺ independent mode and one H⁺ (or OH⁻) dependent). How does this fit with previous knowledge (e.g. genome data) on HCO₃⁻ transporters in diatoms (or *O. sinensis* specifically if available)?
- L. 261-263 How large is the expected effect of a decrease in buffer capacity for the DIC range applied here? Could such an estimate be used to quantify the suggested decrease in eCA activity at low DIC?
- L. 270 What is the specific mechanism of CO₂ supply in the initial phase suggested here (eCA-driven or not)? If the initial CO₂ uptake was not eCA-driven, this should be manifested in a time lag before delta H⁺ starts to drop (in Fig. 4), shouldn't it? Or do you imply a time lag before the inhibitory effect of AZ kicks in (which could be an indication for internalisation of the inhibitor)?
- L. 323 How can the decrease in delta CO₃²⁻ at 8.8 compared to 8.2 be explained? Also, simply judging by eye, this trend (as shown in Fig. 6B) does not seem to be reflected in Fig. 6A – why is this?
- L. 356-359 Do you have data showing this effect (measurements similar to Fig. 1C with an eCA inhibitor)?
- L. 574 At which light intensity and temperature were the measurements performed?
- L. 583 What is the difference between the model used here vs. in Hopkinson et al. 2014?
- L. 592-594 & 180-182 Please clarify whether the values for eCA activity and C fixation are based on measurements in this study (as described in l. 597 ff.) or taken from reference 22. Couldn't the O₂ evolution measurements (l. 567) be used for estimating C fixation?
- What was the average size/dimensions of cells used in the microsensor measurements?
- The manuscript lacks information on the ecological importance of *Odontella sinensis* and any previous knowledge on its CCM (if available) - what is the relevance of these results given the large interspecific variability in CCMs of diatoms? Also, how do the results compare to previous microsensor measurements on diatoms (e.g. Kuhn and Raven 2008)?

Minor comments:

- L. 68 'is may due'?
- L. 117 How would conversion of CO₂ to HCO₃⁻ increase pH? Should this read conversion of HCO₃⁻ to CO₂?
- L. 124 missing 'a'
- L. 147 delete 'at'

- L. 456 What about sinking, could this also have a significant effect on thickness of the boundary layer (and thus applicability of these results to the natural system since there is no flow in the Petri dish)?
- L. 513 By how much did pH vary?
- L. 530 delete 'them'
- L. 572 delete 'at'
- Fig. 1C How do these values relate to those given in lines 146-147?
- Fig. 5A x-axis label is missing
- Is Fig. S4 necessary? To me it looks like Fig. 5A contains essentially the same information.
- Suppl. Fig. S2 (l. 22) 'the pH change itself is not required for the process of photosynthetic carbon uptake': Through which mechanism would carbon uptake require a pH change? To me, the pH change is a consequence rather than a prerequisite for C uptake.

General response

We thank the editor and the reviewers for their helpful and insightful comments. We have attempted to address all of these comprehensively in the revised version of the manuscript. New experimental data is provided to address the issue of diffusion around cells placed on a dish and new cellular modelling data is also provided to address several reviewer concerns. We appreciate that this is a complex subject and have now included a schematic of the major cellular processes (Fig. 2) to ease interpretation of our findings.

A summary of the major changes to figures is listed below.

New Figures

- Supplementary Fig. S6 includes new modelling data illustrating the dynamics of CO_3^{2-} and pH under various bulk seawater pH conditions designed to mimic those measured experimentally in Fig. 6.
- Supplementary Fig. S9 is a new figure incorporating data obtained from cells supported on a fine mesh to allow the diffusive boundary layer to form all around the cell.

Revised Figures

- Fig. 2 now includes a schematic illustrating the major cellular processes involved in the different modes of DIC uptake.
- Fig. 4 has been revised to include multiple traces to illustrate the reproducibility of the data.
- Fig. 6 has been re-drawn to show the changes in $[\text{CO}_3^{2-}]$ at pH 8.8 more clearly.

Response to specific reviewers' comments:

Reviewer #1 (Remarks to the Author):

This manuscript addresses an important topic: the role external carbonic anhydrase in the carbon uptake process in marine diatoms. A strength is the marriage of cutting-edge measurement techniques and a sophisticated model. It is disappointing that the manuscript is marred by numerous typos and errors (some of which I have flagged below). The results are detailed and the manuscript would be more effective if the storyline was strengthened to a crisp message.

Response: We apologise for the typographical errors and have endeavoured to remove all from the revised manuscript. We have substantially trimmed the text and aimed to improve clarity where indicated by the different reviewers.

L43. The issue is not the 1% of CO_2 as DIC but the low (air-equilibrium) concentration relative to Rubisco kinetics. However, the 99% of remaining DIC does represent an opportunity.

Response: The requirement for a CCM is driven by the combination of the availability of CO_2 and the properties of RuBisCO (i.e. low affinity and specificity). We accept the point the reviewer is making and have amended the text to improve clarity (Line 38-41).

L55 – 59 There seem to be some missing statements in the logic here. It would help to be more specific and clarify the link between diffusion limitation and low rate of interconversion of HCO₃⁻ and CO₂.

Response: We have amended the text to describe the modelling work more clearly (L59-61).

L66 This is presumably referring to growth rate.

Response: Yes, growth rate is enhanced. We have clarified this (L72).

L68 Typo.

Response: Corrected.

L87 I am not sure this was one of the conclusions in this paper.

Response: It is indeed one of the key conclusions of the Clement et al 2016 New Phyt paper. They state in the abstract ‘CA was highly and rapidly activated on transfer to low CO₂ and played a key role because inhibition of external CA produced uptake kinetics similar to cells grown at high CO₂.’ However, in the interests of space this reference has been removed from the revised manuscript.

L96 Typo.

Response: Corrected.

L102 What discrepancies are being referred to? Different species may operate differently anyway. There is also a non-sequitur here: different experimental approaches and underestimation of eCA?

Response: We agree that there are different modes of operation of the CCM, but we were referring specifically to the dispute over various proposed roles of eCA in either supplying CO₂ for CO₂ uptake or acting to scavenge CO₂ loss. This text has been revised (L106-109).

L113-114 Only one of the quoted references appears to relate to foraminifera. How photosynthetic are foraminifera? I understand some have endosymbiotic algae. Some references do not refer to single large cells as implied.

Response: Foraminifera themselves are non-photosynthetic but many harbour multiple symbionts (including diatoms). The other references refer to multicellular organisms, as stated. We have rephrased this sentence so that it is clear which references apply to the different statements (L115-117).

L124 Missing word.

Response: Corrected

L127 It would be helpful to be more specific than ‘to increase the supply of CO₂’

Response: We have amended the sentence to clarify that eCA acts to generate CO₂ from HCO₃⁻ at the cell surface (L137-140).

L137 'Significant' in a statistical sense? If not use a different adjective. Check other occasions where 'significant' is used.

Response: The cell surface pH in the light is dramatically different from the dark and it is not necessary to demonstrate this with a statistical test. As we have not applied statistics, we agree that it is better to replace 'significant' with 'substantial' in this sentence (L150).

L139 Decrease and then a -ve value? Would be easier for most readers if [H⁺] was expressed in pH units (or at least a conversion given).

Response: we have removed the -ve sign. Whilst we agree that use of pH units is far easier for most readers to grasp, as pH is a log scale its use creates problems when comparing responses from individual cells. It is far better to express these changes as Δ[H⁺]. To help address this issue, we have provided two axes on the figures to demonstrate how changes in [H⁺] relate to changes in pH. We have also provided a conversion in the text at relevant points to help the reader relate the changes in [H⁺] to pH (L152).

Figure legends- These are quite wordy and descriptive and do not seem to match the journal style.

Response: We have tried to slim down the text of the figure legends where appropriate.

L157 Give pH that the cells were buffered at.

Response: This information is described in the figure legend, but we have now also added it to the text (pH 8.2) (L165).

L173-174 I would have thought the simplest explanation for the pH increase would have been removal of inorganic carbon in photosynthesis and a consequent increase in pH via the buffering in the carbonate system. This might be what is meant by conversion of bicarbonate to CO₂ but it could be more clearly expressed.

Response: The referee is correct that drawdown of CO₂ by carbon fixation will lead to an increase in pH through buffering via the carbonate system and that conversion of HCO₃⁻ to CO₂ is part of this process. We accept that the sentence could be improved and have revised the text (L122).

L182 Does the model assume that eCA is present or effective at distance up to 250 um from the cell surface?

Response: No, the model is spatially explicit and eCA only acts at the cell surface. This is has now been stated in the Methods (L581).

Fig. 2 What does a negative CO₂ concentration mean? I presume it is linked to the assumption of a fixed uptake rate, but good to clarify.

Response: We used a fixed rate of CO₂ uptake to illustrate the problem of limited CO₂ supply, which results in a negative value in the model output. We have amended the figure legend to explain this more clearly.

L192 I have struggled with this and Fig. 2. My first thought was that in the CO₂ uptake simulation, the absence of eCA would generate a higher pH at the cell surface because an out-of-equilibrium situation would have lower CO₂ and higher HCO₃⁻ concentrations than at the new equilibrium. I realise that the model has been described before but it would help clarity and reader comprehension if an additional sentence or two were added in explanation.

Response: We believe the reviewer is considering CO₂ as an acid and thus equating lower CO₂ with a higher pH. This is indeed correct when the system is in equilibrium (when eCA is present), because the uptake of CO₂ results in re-equilibration of the inorganic carbon system with a net consumption of H⁺ through protonation of HCO₃⁻ producing CO₂. However, without eCA these re-equilibration reactions are just too slow and effectively CO₂ is not involved in the acid/base equilibrium. The CO₂/HCO₃⁻ pair could be thought of as “potential” acid/base pair, but because of its slow kinetics the pair is not relevant to the acid/base equilibrium near the cell surface where residence times of chemicals are very short. We have provided a schematic illustration of the major DIC and H⁺ fluxes around the cell to aid with the interpretation of this figure (Fig. 2).

L202- Isn't the distinction between diffusion limitation via CO₂ and HCO₃⁻ uptake not to do with concentration but concentration difference? Even though the diffusion coefficient for HCO₃⁻ is about half that of CO₂ in water, the concentration difference generated by an increase in pH is about 20 times greater for HCO₃⁻ vs CO₂ in your system.

Response: The important point when determining diffusion limitation is the proportion of the pool that is depleted. For similar rates of DIC uptake, the relative depletion of [HCO₃⁻] at the cell surface due to HCO₃⁻ uptake is small (c 5%). In contrast the proportion of the CO₂ pool that is depleted is huge (in fact our model indicates that it would become entirely depleted if CO₂ uptake was able to occur at a fixed rate). We have amended the text to clarify this (L202-204).

L225 Typo.

Response: Corrected

General- Both Fig and Fig. are used. Tidy up.

Response: Corrected to Fig. throughout

L322 Isn't this smaller range simply down to carbonate chemistry?

Response: Yes it is, although we feel it is important to illustrate how changing carbonate chemistry influences processes at the cell surface. In the light of similar comments from reviewer 3, we have now added a comparison between our experimental data and simulations of carbonate chemistry from the cellular model (Supplementary Fig. S6).

General One important point I think not discussed is that although eCA will increase the rate

of CO₂ supply it will not elevate the CO₂ concentration at the cell surface above the bulk-water concentration and is therefore not a CCM and so does not solve the problem of low CO₂ compared to Rubisco kinetics given K_m for diatom Rubiscos of 23 to 68 μM reported in Young et al. (2016) JXB doi:10.1093/jxb/erw163.

Response: This is a good point. eCA itself acts to increase the supply of DIC (in the form of CO₂) but its action alone does not concentrate carbon inside the cell. However, it should be considered a component of the CCM as it allows the cell to utilise diffusive CO₂ uptake across the plasma membrane. We have added a section in the discussion to clarify this issue (L356-360).

L361 is wave focussing likely to act at a relevant timescale?

Response: as we see substantial effects on cell surface pH within seconds, it is possible that wave focusing and flicker light will impact the microenvironment around the cell. The sentence aims to illustrate that phytoplankton do not exist in a stable light environment and that irradiances will likely vary continuously.

L384 Just under or just over? (57%).

Response: We have replaced 'just under' with 'approximately' to correct the error and avoid further confusion (L388).

L575 Temperature monitored but it has not been stated explicitly what the temperature was (presumably the 15oC used for growth, but the model and MIMS at 20oC).

Response: The cells were grown at 15°C, but for practical reasons all experimental manipulations were performed at 20°C. Therefore the model simulates 20°C. We have clarified this in the Methods.

The references need tidying up.

Response: We have carefully checked the references and corrected any errors.

Reviewer #2 (Remarks to the Author):

The paper describes experiments with O₂, pH and CO₃²⁻ microsensors on diatoms. Especially the effects of CA inhibitors was tested, for external CA.

Inhibition of CA resulted in reduction of photosynthesis, pH dynamics and CO₃²⁻ dynamics. The inhibitor acts directly or indirectly on photosynthesis, and reduces CO₂ fixation. How this functions is the subject of this paper.

I have not the feeling it is truly understood, that we have progressed our insights in the question why CA activity is needed for DIC uptake. It is a complex issue, where it is easy to get confused. I became confused and still do not know how exactly CA is involved in DIC uptake.

Response: We agree that it is a complex and potentially confusing issue. By making direct measurements of the microenvironment, we can avoid making assumptions about the processes occurring at the cell surface and provide new information for many other researchers in this field. The carbonate chemistry at the cell surface is determined by a combination of many processes, each occurring at different rates. These are primarily diffusion, interconversion between the different forms of DIC and fluxes of molecules across the plasma membrane. To understand the net outcome of all these processes and to help us interpret our measurements requires an advanced cellular model. Our findings demonstrate that diffusion limitation of CO₂ does indeed occur around large diatom cells and that eCA plays an important role in maintaining [CO₂] at the cell surface. Detailed answers to specific points raised are provided below. In order to clarify the nature of the interactions further we have provided a schematic illustrating the processes associated with different modes of DIC uptake (Fig. 2).

The aim of the measurements should be explicitly formulated. A hypothesis and its test is missing.

Response: The hypothesis that we test is derived from prior modelling studies (e.g. Wolf-Gladrow and Riebesell 1997 *Marine Chemistry*) that predict that CO₂ supply to large diatoms may be limited by their large diffusive boundary layer. We test this experimentally and demonstrate that eCA is essential to maintain [CO₂] at the cell surface. We have now stated this explicitly in the Introduction (L128-130).

Also, it may be better explained what the purpose of carbonate sensors is.

Response: The purpose of the CO₃²⁻ sensors is to allow us to more fully understand carbonate chemistry around the cell. By using simultaneous measurements of both pH and [CO₃²⁻] we are able to detect cellular activities that would not be apparent from measurements of pH alone. An example of this is the decrease in [CO₃²⁻] observed following inhibition of eCA (Fig. 5A), which we propose is indicative of active HCO₃⁻ uptake. We have added a sentence to help clarify this to the reader (L283-285).

It is really interesting that on single cells the surface dynamics are as large as they are recorded here. The work is technically extremely challenging and seems very well done. Some interesting observations are made, such as the O₂ peak in Fig. 4, but we need insight in how reproducible the results are. It must be mentioned how many measurements are made.

Response: In each case we have stated the number of cells examined (e.g. in Fig. 4 it is stated that n=7 cells were examined). There is an important balance when performing single cell analyses between showing the intricate detail of a single cell's response and demonstrating the reproducibility between cells. In this case (Fig. 4) we showed a representative trace as showing all 7 traces made it difficult to discern the effect clearly. We have now revised the figure to include multiple traces and clarified the sample number in the text (L275).

DIC is taken up as CO₂ and bicarbonate. The CO₂ supply is rapidly exhausted as the pool is small, thus CA has a role in converting bicarbonate into CO₂. CA inhibitors will thus reduce CO₂ uptake, which could be compensated by bicarbonate uptake. Clearly this compensation does not happen, as bicarbonate is in equilibrium with carbonate, and carbonate dynamics are reduced by CA inhibition.

Response: Our results show some evidence for compensation. The decrease in $[\text{CO}_3^{2-}]$ observed when eCA is inhibited suggests that HCO_3^- uptake is activated. This is because HCO_3^- uptake will result in a decrease in $[\text{HCO}_3^-]$ at the cell surface and a subsequent decrease in $[\text{CO}_3^{2-}]$ as it rapidly equilibrates with HCO_3^- . These observations are supported by our cellular model (Fig 3C) and are indicative of a degree of compensation. However, the degree of compensation is only slight as O_2 evolution is still only 50% of a cell with active eCA.

Thus CA is also essential in bicarbonate uptake, or the inhibitor inhibits the bicarbonate uptake.

Response: We strongly disagree with this statement and do not follow the logic of this argument. Inhibition of eCA results in substantial reduction in O_2 evolution. The simplest explanation is that *O. sinensis* uses a combination of CO_2 and HCO_3^- uptake, with eCA acting to support CO_2 uptake. Following the addition of AZ, CO_2 uptake is therefore greatly reduced but HCO_3^- uptake continues at a similar rate (or is stimulated slightly) to support the remaining 50% of O_2 evolution. There is no evidence to suggest that AZ has direct impact on HCO_3^- uptake (see response below).

The question is now: how does AZ inhibit photosynthesis? Via DIC uptake most likely, but exactly how? Why is CA, thus the hydration of CO_2 to bicarbonate, important for DIC uptake?

Response: It is important to clarify the terms used here. DIC refers to all carbonate species (CO_2 , HCO_3^- and CO_3^{2-}). Our data show that eCA is important for eCA-catalysed CO_2 uptake, but we find no evidence for a requirement for eCA in HCO_3^- uptake. If the primary function of eCA was to catalyse the hydration of CO_2 to HCO_3^- , as suggested by the referee, then its activity would be accompanied by the production of H^+ at the cell surface and a substantial decrease in pH. Our measurements clearly indicate that this does not occur.

Could AZ have a direct effect on the bicarbonate uptake? That would be the easiest explanation, but it does not provide insight in the processes. The authors have not been able to find the role of CA in DIC uptake.

Response: We address this concern in the Discussion (L419-425). Direct measurements show that AZ has no effect on the SLC4 bicarbonate transporter of *Phaeodactylum*, even at 100 μM (Nakajima 2013 PNAS). Moreover, our data suggest a stimulation rather than inhibition of HCO_3^- uptake in the presence of eCA inhibitors.

We do not follow the argument that we have been unable to find the role of eCA in DIC uptake as we provide strong evidence to indicate that the cell uses eCA to maintain CO_2 at the cell surface.

It seems to me that the data are a bit over-interpreted. I do not follow the note that pH and CO_3^{2-} dynamics are driven by the activity of CA. They are driven by photosynthesis. CA only brings the carbonate system, incl. pH, closer to equilibrium.

Response: We agree that it is ultimately photosynthesis that dictates the DIC fluxes and that the role of eCA is catalyse the equilibrium between CO_2 and HCO_3^- . Therefore the term 'driven' is perhaps inappropriate for eCA activity and we have removed it from the text.

However, it is clear that when eCA is inhibited the changes in pH and $[\text{CO}_3^{2-}]$ are greatly reduced, whilst photosynthesis still occurs at approximately 50% of the control. This demonstrates that the activity of eCA, rather than CO_2 fixation, is primarily responsible for the changes in cell surface carbonate chemistry.

In some occasions the authors used H^+ concentrations, in others pH, so $-\log[\text{H}^+]$. Essentially the same but written differently. Stick to one notation, preferably pH.

Response: As discussed in the response to referee 1, it is problematic to use pH to calculate the mean response to a stimulus as pH is a log scale. Therefore we have used $[\text{H}^+]$ for the majority of comparisons between cells. However, we agree that most readers will relate to pH much more easily than $[\text{H}^+]$. Therefore, whilst we have used $[\text{H}^+]$ for all comparisons, we have indicated both $[\text{H}^+]$ and pH on figures and provided a conversion to pH where appropriate. See also response to reviewer 1.

On several occasions an increase in CO_2 was reported at the surface of phototrophs upon CA inhibition. This could be part of the discussion.

Response: We are unsure precisely which studies the referee is referring to and we are not aware of any direct measurements of $[\text{CO}_2]$ at the surface of algal cells. However, there is evidence that organisms that rely on active HCO_3^- uptake accumulate CO_2 outside the cell. This has been demonstrated primarily in cyanobacteria (e.g. Tchernov 1997 Current Biology), which lack eCA and actively pump HCO_3^- into the cell and then convert it to CO_2 at the site of carbon fixation (in the carboxysome). The accumulated CO_2 inside the cell can leak out, causing a rise in $[\text{CO}_2]$ outside of the cell. This effect has also been observed in some eukaryotes that lack eCA (e.g. *Nannochloropsis*), but is not found in marine diatoms that possess eCA (Tchernov 1997). It seems that elevated $[\text{CO}_2]$ outside the cell is indicative of CO_2 leakage during active HCO_3^- uptake, but this would not be observed in organisms that rely on diffusive CO_2 uptake. We appreciate that it would be useful to mention this in the Discussion and have added an appropriate citation (L344-346).

The discussion is rather unstructured and too long. The discussion should better focus on the role of CA in DIC uptake and explain why AZ reduces DIC uptake and photosynthesis. The thinking about irregular shapes and fluctuations is not essential for the concept. The methodological section and the carbonate environment are superfluous.

Response: We have added text to clarify the role of eCA (L356-360) and have substantially trimmed the Discussion to improve focus. However, it is also clear from the comments of the referees that some of the sections suggested to be trimmed are useful in the interpretation of the data. For example, the concerns of the referee addressing inhibition of HCO_3^- uptake by AZ and the purpose of the carbonate sensors (see above) are both dealt with in the Discussion (L419-425; L426-434).

L132 the MBL does not directly influence processes, but processes together with the MBL change concentrations at the cell surface.

Response: We have clarified the sentence (L143).

L207 This seems a crucial section. The CO_2 is depleted due to the pH shift? eCA activity during photosynthesis leads to CO_2 depletion due to the pH shift and ensuing shift in

carbonate equilibrium towards bicarbonate. It may indeed also be that the bicarbonate depletion by uptake leads to CO₂ conversion to bicarbonate and a pH increase. Inhibiting CA will then not lead to less DIC uptake (as bicarbonate is not limiting) and hence to less photosynthesis.

Response: This is indeed a crucial point, but it should be made clear that this statement (L210-212) refers specifically to the theoretical effect of expressing eCA during HCO₃⁻ uptake, as explored by the cell model. The major point is to illustrate that inhibition of eCA has a very different impact on cell surface pH and [CO₃²⁻] depending on whether the cell is taking up CO₂ or HCO₃⁻ and that we can then test this with our experimental data. As the referee suggests, the depletion of [CO₂] by eCA during HCO₃⁻ uptake is a combination of the effect of increasing pH and depletion of [HCO₃⁻] at the cell surface (this can be clearly observed in Fig 3C where we vary the rate of H⁺ uptake/OH⁻ extrusion).

The referee is correct in stating that depletion of [HCO₃⁻] would lead to eCA-catalysed CO₂ conversion to HCO₃⁻. However, this would result in a pH decrease not an increase as stated by the referee. This pH effect can be observed in our cellular model (Fig 2H, compare solid vs dotted line), although it is only a minor influence compared to the impact of H⁺ uptake. We agree with the referee that inhibition of eCA in a cell relying on HCO₃⁻ uptake will not negatively affect DIC uptake or photosynthesis. We have added a schematic to improve clarity (Fig 2).

L244 the pH dynamics are lower, as P is lower. How DIC is taken up can indeed not change the pH at the surface as the pH is in the end controlled by the net CO₂ fixation rate, regardless of the mechanism of DIC uptake.

Response: We disagree with this comment. There is an important distinction between the effect of CO₂ fixation on the bulk seawater and its impact on the microenvironment. The pH at the cell surface of a photosynthetic organism is controlled by the combination of the net fluxes of H⁺ and other ions across the plasma membrane, by the balance of H⁺ consuming and H⁺ producing reactions in the DBL and by diffusion of H⁺ and other ions to and from the cell surface. This is most apparent in the giant internodal cells of the alga *Chara*, which possess distinct acid and alkali zones at the surface due to spatial separation of membrane transport processes during photosynthesis (e.g. Krupenina et al Photochem. Photobiol. Sci., 2008, 7, 681–688). Although *O. sinensis* cells are orders of magnitude smaller than *Chara*, we demonstrate that they also show a non-uniform cell surface pH. These examples very clearly illustrate that cell surface carbonate chemistry is not solely a direct reflection of the rate of CO₂ fixation and is determined by the specific reaction/diffusion environment at the cell surface and the underlying membrane transport processes. Whilst a change in net CO₂ fixation rate will of course have a major impact on cell surface pH, this effect will be different depending on the mechanism of DIC uptake (e.g. active HCO₃⁻ uptake vs diffusive CO₂ entry), which result in different reactions and fluxes at the cell surface.

L263-271 A bit too detailed info. How often have these experiments been repeated? How significant are the data?

Response: Our intention was only to state that an increase in the inhibition of O₂ evolution by AZ was not observed. We have now modified the text to state this clearly. The transient peak in cell surface [O₂] was observed in 7 out of 7 cells examined at 0.5 mM DIC, whereas it was absent in all of the other analyses performed at 2 mM DIC throughout this study. We have included this observation as we believe it to be informative to the reader, but we agree

that observations from single cells are subject to variability and have not tried to over-interpret the data. The replication has been stated in the figure legends, but we have now included it in the text to increase clarity (L266-267).

The peak is due to a rapidly disappearing stored pool of DIC. That pool should also be there at 2 mM DIC. Why does the O₂ level increase again after t=100 s?

Response: As the transient [O₂] peak occurs after inhibition of eCA (i.e. external CA), we feel that it most likely reflects the reduced availability of external CO₂ at low DIC, rather than changes in an internal DIC pool. In the initial period following 'light on', photosynthesis is likely to be supported by both CO₂ and HCO₃⁻ uptake, whereas O₂ evolution after this period is primarily supported by HCO₃⁻ uptake only. The transient decrease in [O₂] therefore reflects a decrease in the rate of DIC uptake as the cell rapidly switches from CO₂ + HCO₃⁻ uptake to HCO₃⁻ uptake only.

It is better to provide the absolute O₂ concentrations instead of % of the untreated control. What is the untreated control? Clearly 100% is not the sample without AZ (the solid line), as that varies as well. So what is 100%?

Response: We show the absolute changes in [O₂] around a single cell in Fig 1A and we agree that it would be preferable to report absolute [O₂] throughout. However, we felt that for Fig 4 the variability between cells would give a misleading impression. As detailed in the methods, the O₂ microsensor was much larger than those used for pH and [CO₃²⁻] measurements and we found that decreased spatial resolution led to variability in our measurements of the change in [O₂] between cells. This was not a problem when determining the relative effects of an inhibitor on a single cell, but presents an issue when comparing data between cells. All O₂ measurements were therefore derived from the relative effects of a treatment on individual cells and were presented as % of the increase in the untreated control (i.e. without inhibitor) rather than absolute rates. 100% was defined as the maximum [O₂] reached in the untreated control (i.e. without AZ) above the initial value. We have clarified this in the figure legends.

L296 This should be mentioned earlier: carbonate dynamics provide direct info in the bicarbonate uptake. An advantage over CO₂ microsensors, as CO₂ is not closely coupled to the DIC system.

Response: We should point out that CO₂ is very closely coupled to the DIC system, as DIC is composed of CO₂, HCO₃⁻ and CO₃²⁻. We think that the referee is referring to the fact that CO₃²⁻ sensors allow insight into HCO₃⁻ dynamics as the equilibration between HCO₃⁻ and CO₃²⁻ is very rapid, whereas the equilibration between CO₂ and HCO₃⁻ is much slower (hence the need for carbonic anhydrases). We agree that it would be useful to highlight this in the manuscript and have added a line to that effect (L283-285).

L341-345 contradictory. If bicarbonate is taken up the equilibrium will shift from CO₂ to bicarbonate leading to pH increase. How can CA prevent CO₂ loss? Loss where to and from where? This concept must be better introduced, discussed and explained.

Response: The statement is not contradictory. The proposed conversion of CO₂ to HCO₃⁻ will result in a pH decrease, not an increase as suggested by the referee (according to the equation CO₂ + H₂O → HCO₃⁻ + H⁺). The proposed role for eCA in preventing CO₂ loss has been explained in detail in the cited references (e.g. Trimborn 2009; Trimborn 2008; Martin

and Tortell 2008) and this concept is explained in the Introduction. We have amended the text in the Introduction to help clarify this concept for the reader (L98-101) and have also illustrated this proposed process in a schematic figure (Fig. 2).

If CA is inhibited CO₂ increases outside phototrophs.

Response: Although we are unsure exactly which research the referee is referring to, we assume that this comment refers to the increase in extracellular [CO₂] observed in cyanobacteria, which is caused by CO₂ leakage following active HCO₃⁻ uptake. We have provided a detailed response to this in a previous comment. We would not expect net CO₂ leakage when a cell is primarily using CO₂ uptake for photosynthesis.

L344 Please describe the process that leads to the pH excursion. CO₂ fixation leads to the pH increase. CA seems rather responsible for the DIC supply, as inhibiting it reduces the photosynthesis strongly. pH increase alone does not normally do that.

Response: As stated above, it is important to distinguish between the effect of CO₂ fixation on the bulk seawater and its impact on the cellular microenvironment. At high cell densities, CO₂ fixation can result in net decrease in DIC of the bulk seawater and an increase in pH as equilibration with atmospheric CO₂ does not occur sufficiently rapidly. However, carbonate chemistry at the cell surface is subject to spatial constraints (e.g. diffusion limitation) and can therefore differ considerably from the bulk seawater. Therefore, whilst the rate of CO₂ fixation plays a critical role in defining the magnitude of these fluxes, carbonate chemistry at the cell surface is defined by the many processes occurring at the cell surface, which differ depending on the mode of DIC uptake.

The increase in cell surface pH is due to the activity of eCA in converting HCO₃⁻ to CO₂. Our model shows that uncatalysed conversion of HCO₃⁻ to CO₂ is too slow to result in an appreciable change in pH. We agree that eCA seems at least partly responsible for DIC supply, as inhibiting eCA leads to a substantial reduction in the rate of photosynthetic O₂ evolution.

L366 means that CO₂ in the cells is lower than outside. This is unlikely and would seriously make Rubisco ineffective. Kaplan and others have shown that CO₂ is higher inside.

Response: CO₂ is not transported actively across cellular membranes and so net fluxes are therefore determined by concentration gradients. In order to take up CO₂ across the plasma membrane, [CO₂] in the cytosol must be lower than that at the external cell surface. However, it is important to remember that [CO₂] in the cytosol does not reflect [CO₂] at the site of carbon fixation, which occurs inside the chloroplast. Diatoms actively accumulate DIC in the chloroplast (either through active transport of HCO₃⁻ into the chloroplast or through the action of a C₄ biochemical CCM), where intracellular CAs (iCA) enable the generation of elevated [CO₂] for fixation by RuBisCO. This mechanism (passive diffusion of CO₂ across the plasma membrane, followed by active accumulation of DIC into the chloroplast) requires a net inward CO₂ gradient across the plasma membrane. eCA is therefore required to maintain the [CO₂] at the cell surface. We appreciate that the different processes described in the manuscript are complex and we have included a schematic model to improve clarity (Fig. 2).

These processes are described in detail in a number of recent reviews on the operation of diatom CCMs (e.g. Hopkinson et al 2016, 3, p51-57, Current Opinion in Plant Biology). A figure from this review is shown below to aid the review process.

We presume that the referee is referring to the intracellular accumulation of CO_2 demonstrated by Kaplan and co-workers in cyanobacteria, which of course do not have chloroplasts and rely primarily on active transport of HCO_3^- into the cell. As pointed out by referee 1, eCA does not act to concentrate carbon inside the cell, but contributes to the activity of the CCM by facilitating diffusive uptake of CO_2 across the plasma membrane.

What is the flexible response by CA to fluctuations?

Response: By expressing eCA, the cell can maintain a near constant $[\text{CO}_2]$ at the cell surface, even if the cell has an irregular shape or light intensity is erratic. We have rephrased the text to avoid confusion (L370-372).

Reviewer #3 (Remarks to the Author):

General comments:

*This manuscript by Chrachri et al. describes carbonate chemistry at the surface of single diatom cells based on microsensor measurements of pH, CO_3^{2-} and O_2 . The results are further analyzed with a model of cellular carbon fluxes to infer information on the mode of carbon uptake, concluding that eCA plays an important role in carbon uptake in *O. sinensis* by accelerating CO_2 supply in the boundary layer.*

Overall, it is a well written, convincing manuscript that provides important new insights going beyond a mere description of boundary layer carbonate chemistry. By combining these measurements with very instructive inhibitor experiments and model calculations, the authors are able to significantly advance our understanding of carbon uptake mechanisms in diatoms. However, I have one major question/concern regarding the methodology which needs to be clarified, as well as several more specific comments and questions:

- Regarding the methodology, I wonder how placing the cells on the solid bottom of a Petri dish affects concentrations of O_2 and H^+ in the boundary layer as compared to a free-floating cell. I am concerned that by preventing diffusion to/from below the cell, the glass bottom may distort concentrations at the cell surface, gradients in the boundary layer (Fig. 1D) and photosynthesis estimates by the Revsbech et al. approach, as demonstrated for phytoplankton aggregates (Ploug and Jorgensen 1999 MEPS). This depends on diffusion within/through the cell which is difficult to estimate, but the magnitude of this effect could be tested e.g. by

comparing gradients between cells placed on agar vs the solid glass surface (as the diffusion coefficient for O₂ is the same in 1% agar as in water, see e.g. Ploug et al. 2010, ISME J).

Response: Firstly, it is important to note that whilst the dish could potentially affect the size and shape of the diffusion boundary layer, it would not alter the underlying cellular mechanisms, which is the major focus of our manuscript. Moreover, the phytoplankton aggregates measured by Ploug and Jorgensen (1999) are several orders of magnitude larger than single phytoplankton cells, with a radius of several millimetres. Therefore, we believe that any dish effects are unlikely to affect our major conclusions. However, the reviewer has highlighted a valid concern and so we have addressed this issue with additional experiments. We measured cell surface pH around *O. sinensis* cells suspended on a fine nylon mesh (pore size 100 µM) to allow free diffusion around the cell from all directions. We still observed significant increases in pH upon illumination, which were comparable to those observed on a dish. Therefore, any effect of the dish on the size and shape of diffusion boundary layer is unlikely to influence our conclusions. We have included these data as a new supplementary figure (Supplementary Fig. S9).

- I find it surprising that under dark conditions, pH and O₂ at the cell surface were virtually the same as in the bulk (l. 134-136). Furthermore, it is interesting that O₂ concentrations decrease below air saturation in the dark when an eCA inhibitor is present (Fig. 3A). How could this be explained?

Response: We agree that this is slightly surprising as measurements around larger cells (e.g. foraminifera) indicate a significant decrease in pH associated with respiration. We have therefore included additional data to address the reviewer's concerns. There is a noticeable decrease in cell surface pH relative to the bulk seawater in the dark, but the amplitude of this change is much smaller than the increase in pH observed in the light. We have included this information in the revised text (L144-146). We have also included an example from a suspended cell in Supplementary Fig. S9.

We believe the lower [O₂] observed in the presence of AZ seen in Fig. 3C may be primarily due to the time it takes for O₂ to reach equilibrium after each light/dark transition. The much greater [O₂] reached in the absence of AZ means that [O₂] probably has not fallen to its true dark level after each 300s dark period. However, we cannot rule out a physiological contribution to this effect, as it is clear that [O₂] does not simply decrease exponentially during the dark period for the cell illustrated. This was not observed for every cell (e.g. Fig. 1), so we do not have an obvious explanation for this effect. Detailed examination of the kinetics of the pH/O₂ changes will clearly be an interesting area for further study.

- L. 245 HCO₃⁻ uptake in O. sinensis is suggested here to consist of two different uptake modes (one H⁺ independent mode and one H⁺ (or OH⁻) dependent). How does this fit with previous knowledge (e.g. genome data) on HCO₃⁻ transporters in diatoms (or O. sinensis specifically if available)?

Response: We refer to the requirement to balance HCO₃⁻ uptake with H⁺ uptake (or OH⁻ efflux) for the intracellular generation of CO₂, rather than the specific mechanism of HCO₃⁻ uptake. Knowledge of the actual mechanisms of HCO₃⁻ uptake are limited, but it is clear that marine diatoms possess members of the SLC4 and SLC26 family of anion exchangers. Whilst it is difficult to predict the substrate specificity of these proteins from sequence information alone, characterisation of the SLC4-2 transporter from *Phaeodactylum* revealed it to be a

Na⁺-coupled transporter (Nakajima 2013 PNAS). We have clarified the text in the introduction and results to avoid confusion (L54-57, 199-201).

- L. 261-263 *How large is the expected effect of a decrease in buffer capacity for the DIC range applied here? Could such an estimate be used to quantify the suggested decrease in eCA activity at low DIC?*

Response: We agree that modelling the effect of 0.5 mM DIC on cell surface carbonate chemistry would be useful. We therefore ran the model at 2 mM DIC and 0.5 mM DIC assuming that DIC uptake is split 50% between HCO₃⁻ uptake and CO₂ uptake. The results indicate that a larger pH increase will be observed at 0.5 mM DIC, suggesting that the experimental data are most likely explained by the decrease in buffer capacity (L266-268). Whilst the model is extremely useful for examining expected trends in the experimental data, there are many assumptions in the model (mode of DIC uptake, rate of photosynthesis, size and shape of cell) that make it tricky to predict precisely what we might observe experimentally. We are therefore cautious about using this estimate to directly gauge whether there has been any reduction in eCA activity in the experimental data.

- L. 270 *What is the specific mechanism of CO₂ supply in the initial phase suggested here (eCA-driven or not)? If the initial CO₂ uptake was not eCA-driven, this should be manifested in a time lag before delta H⁺ starts to drop (in Fig. 4), shouldn't it? Or do you imply a time lag before the inhibitory effect of AZ kicks in (which could be an indication for internalisation of the inhibitor)?*

Response: It is clear from our perfusion experiments that inhibition of eCA by AZ is very rapid (e.g. Fig 3A, Fig 5A) and we observe no progressive decline in O₂ evolution that would be indicative of internalisation of the inhibitor (i.e. due to inhibition of iCAs). Therefore, we expect that eCA is fully inhibited in the initial period and we agree that cell surface pH should therefore not change substantially in the absence of eCA activity. This is exactly what we observe (Fig 4 lower right panel, dotted line), suggesting that the initial phase of O₂ evolution is not supported by eCA activity.

- L. 323 *How can the decrease in delta CO₃²⁻ at 8.8 compared to 8.2 be explained? Also, simply judging by eye, this trend (as shown in Fig. 6B) does not seem to be reflected in Fig. 6A – why is this?*

Response: The decrease in delta [CO₃²⁻] at 8.8 relative to 8.2 is most simply explained by differing chemical response of the inorganic carbon system at the two pHs. We modelled the effects of photosynthesis and DIC uptake on pH and CO₃²⁻ at the cell surface at different bulk pHs. Photosynthetic and DIC uptake rates were kept constant in all simulations. The modelled changes are in good agreement with the observed changes and in particular the change in [CO₃²⁻] is less at pH 8.8 than pH 8.2 (Supplementary Fig. S6, L324-328). Despite the higher [CO₃²⁻] at pH 8.8, the surface pH changes only slightly, leading to a reduced change in cell surface [CO₃²⁻] at pH 8.8 compared to 8.2.

We agree that the trend shown in Fig. 6B is not clearly reflected in Fig. 6A. We have redrawn Fig 6A to illustrate this trend more clearly.

- L. 356-359 *Do you have data showing this effect (measurements similar to Fig. 1C with an eCA inhibitor)*

Response: Unfortunately, we have been unable to acquire data to show this effect. In the absence of eCA, depletion of [CO₂] around the cell has very little impact on the other components of the carbonate system (Fig 2). Therefore to show that there were different levels of CO₂ around the cell would require extensive development of a CO₂ microsensor with the necessary spatial resolution. Whilst this is technically feasible, it is beyond the scope of our current study.

- L. 574 *At which light intensity and temperature were the measurements performed?*

Response: The light intensity was 200 μmol m² s⁻¹ unless otherwise stated and the temperature was 20°C. This information has been added to the Methods.

- L. 583 *What is the difference between the model used here vs. in Hopkinson et al. 2014?*

Response: The model is very similar to the spherical reaction diffusion model used in Hopkinson et al (2013) *Plant Physiology*. Input parameters such as cell size and photosynthetic rate were changed for this study (see below) but the model itself was identical.

- L. 592-594 & 180-182 *Please clarify whether the values for eCA activity and C fixation are based on measurements in this study (as described in l. 597 ff.) or taken from reference 22. Couldn't the O₂ evolution measurements (l. 567) be used for estimating C fixation?*

Response: We apologise for not making this clear. The model was parameterised using values measured in *O. sinensis* where possible. eCA activity in *Odontella* was measured using MIMS data. The base photosynthetic rate was not measured, but was estimated based on observed pH changes in the absence of an eCA inhibitor. The estimated rate was similar to measured photosynthetic rates of large diatoms in ref 22 (Shen 2015).

Our microsensor O₂ measurements were used to estimate relative photosynthetic rates. In theory, they could be used to quantify O₂ evolution per cell, although as the cell is not spherical and the spatial resolution of the O₂ sensor used in our studies (50 μm) is relatively low (compared to the microelectrode measurements), we felt that these measurements were not ideal for the quantitative determination of photosynthetic rate required for incorporation into the cellular model.

- *What was the average size/dimensions of cells used in the microsensor measurements?*

Response: The *O. sinensis* cells used were between 150-250 μm in length. This information is included in the figure legend of Fig 1.

- *The manuscript lacks information on the ecological importance of *Odontella sinensis* and any previous knowledge on its CCM (if available) - what is the relevance of these results given the large interspecific variability in CCMs of diatoms?*

Response: *Odontella sinensis* is a common large diatom in European waters (Gomez 2010, Widdicombe 2010). We are not aware of previous reports on the functions of its CCM, although the eCA activity exhibited by *O. sinensis* is of a similar level to that found in other large centric diatoms. Whilst eCA can have an irregular distribution in other phytoplankton taxa, it appears to be ubiquitous in centric diatoms, and so we believe that the results are of

broad relevance to the understanding of CCMs in this group. We have amended the text to include this information (L131-132, L375-376).

Also, how do the results compare to previous microsensor measurements on diatoms (e.g. Kuhn and Raven 2008)?

Response: Kuhn and Raven (2008) showed that pH increases substantially at the cell surface of *Coscinodiscus walesii* in the light and that there was little change in cell surface pH from the bulk seawater in the dark, although they did not examine the underlying cellular mechanisms. Their findings are similar to what we observe in *O. sinensis*. Although we cited the reference in the original manuscript, we have now amended the text to clarify this (L117-119).

Minor comments:

- L. 68 'is may due'?

Response: Corrected

- L. 117 How would conversion of CO₂ to HCO₃⁻ increase pH? Should this read conversion of HCO₃⁻ to CO₂?

Response: Yes, we apologise for the error.

- L. 124 missing 'a'

Response: Corrected

- L. 147 delete 'at'

Response: Corrected

- L. 456 What about sinking, could this also have a significant effect on thickness of the boundary layer (and thus applicability of these results to the natural system since there is no flow in the Petri dish)?

Response: It should be noted that all experimental analyses were performed under constant perfusion at a flow rate of 1 mL min⁻¹ (L561-562), which would lead to background level of turbulence in the dish. Although we did not try to quantify the resultant turbulence at a single cell level (and it would be difficult to do this without significant modification of the experimental setup), it is clear that the formation of a diffusion boundary layer occurs even in a constant flow.

We agree that sinking would have an effect akin to turbulence i.e. that it will increase diffusive supply of CO₂, and we have now mentioned sinking in the Discussion (L471-473). Sinking (and turbulent mixing) are unlikely to remove the need for eCA altogether, as our models indicate a continued requirement for eCA even at pH 7.6 where [CO₂] is 3x higher (Supplementary Fig. S6). Clearly, natural variables (light, turbulence, sinking etc) will all influence the extent of the DBL and determining these will be an important area for further research. However, our current manuscript demonstrates how eCA will allow the cell to

maintain an inward diffusive CO₂ gradient in response to these changes in the supply and demand of CO₂.

- L. 513 *By how much did pH vary?*

Response: The pH in culture vessels was maintained between 8.1 and 8.3. Cells were sub-cultured if pH rose above these values.

- L. 530 *delete 'them'*

Response: corrected

- L. 572 *delete 'at'*

Response: corrected

- *Fig. 1C How do these values relate to those given in lines 146-147?*

Response: We erroneously included more replicates in Figure 1C (n=12) than in the text (n=8). We have updated the text to include the additional data (L159).

- *Fig. 5A x-axis label is missing*

Response: corrected

- *Is Fig. S4 necessary? To me it looks like Fig. 5A contains essentially the same information.*

Response: Fig S4 contains similar information to Fig 5A, although we feel that it is important to retain Fig S4 for the flow of the manuscript as Fig 5A is not referred to in the text until much later. Fig S4 also serves to illustrate the reproducibility of the benzolamide response.

- *Suppl. Fig. S2 (l. 22) 'the pH change itself is not required for the process of photosynthetic carbon uptake': Through which mechanism would carbon uptake require a pH change? To me, the pH change is a consequence rather than a prerequisite for C uptake.*

Response: We agree that is unlikely that a pH change would be required for carbon uptake and it is difficult to envisage a mechanism through which this happens. As localised pH changes are important for a range of transport processes in plants and algae, we wanted to illustrate that we have tested this possibility experimentally. However, to avoid confusion we have deleted this phrase.

REVIEWERS' COMMENTS:

Reviewer #2 (Remarks to the Author):

The authors have responded adequately to the reviews. It is a very well written and important paper. The study leads to better understanding of the concepts important in DIC uptake.

Some minor issues:

L523 State what reference was used, probably Ag/AgCl.

L530 the response of the carbonate microelectrodes must have been log-linear, not linear, to concentration.

Reviewer #3 (Remarks to the Author):

Comments on revised version NCOMMS-17-10972A

I think the manuscript has gained clarity through rephrasing of some passages in the text, and the revisions of the figures and the additional supplementary figures are helpful. My comments have been adequately addressed, yet I have one follow-up comment on the discussion of the additional model run made by the authors (referring to the following comment I made on the first version):

L. 261-263 How large is the expected effect of a decrease in buffer capacity for the DIC range applied here? Could such an estimate be used to quantify the suggested decrease in eCA activity at low DIC?

Response: We agree that modelling the effect of 0.5 mM DIC on cell surface carbonate chemistry would be useful. We therefore ran the model at 2 mM DIC and 0.5 mM DIC assuming that DIC uptake is split 50% between HCO₃⁻ uptake and CO₂ uptake. The results

indicate that a larger pH increase will be observed at 0.5 mM DIC, suggesting that the experimental data are most likely explained by the decrease in buffer capacity (L266-268).

Whilst the model is extremely useful for examining expected trends in the experimental data, there are many assumptions in the model (mode of DIC uptake, rate of photosynthesis, size and shape of cell) that make it tricky to predict precisely what we might observe experimentally. We are therefore cautious about using this estimate to directly gauge whether

there has been any reduction in eCA activity in the experimental data.

Follow-up comment: I appreciate the additional model run which is certainly useful, yet I think the discussion of the different DIC treatments still requires some clarification. I assume the low DIC scenario (0.5 mM DIC) yields stronger pH variations because also alkalinity is lower in this scenario compared to the 2 mM standard scenario. This might be meant by 'lower buffer capacity', but it should be clarified in the manuscript to avoid confusion. Note that the decrease in the Revelle factor - which can also be referred to as buffer capacity - at elevated DIC has the opposite effect. This is actually shown in the 'high CO₂ experiment' in Fig. 6 and S6, where the low pH treatment (which must have higher DIC concentrations since it was achieved by bubbling with CO₂ as far as I understand) lead to stronger proton variations, whereas in the example discussed in the response to my question (0.5 mM vs 2 mM DIC), the low DIC treatment lead to stronger proton variations. I think a note on bulk carbonate chemistry (incl. alkalinity) in the different media prepared for the experiment and assumed in the model as well as a definition of the term buffer capacity would help avoid confusion here.

Response to Reviewer's comments (NCOMMS-17-10972A)

Reviewer #2 (Remarks to the Author):

The authors have responded adequately to the reviews. It is a very well written and important paper. The study leads to better understanding of the concepts important in DIC uptake.

Some minor issues:

L523 State what reference was used, probably Ag/AgCl.

Response: The reference electrode was a KCl-filled glass capillary containing a Ag/AgCl wire. We have amended the text to clarify this (ln 528).

L530 the response of the carbonate microelectrodes must have been log-linear, not linear, to concentration.

Response: Yes, the response was linear with the log change of $[\text{CO}_3^{2-}]$, so log-linear is correct. We have changed the manuscript accordingly. (ln 534).

Reviewer #3 (Remarks to the Author):

Comments on revised version NCOMMS-17-10972A

I think the manuscript has gained clarity through rephrasing of some passages in the text, and the revisions of the figures and the additional supplementary figures are helpful. My comments have been adequately addressed, yet I have one follow-up comment on the discussion of the additional model run made by the authors (referring to the following comment I made on the first version):

L. 261-263 How large is the expected effect of a decrease in buffer capacity for the DIC range applied here? Could such an estimate be used to quantify the suggested decrease in eCA activity at low DIC?

Response: We agree that modelling the effect of 0.5 mM DIC on cell surface carbonate chemistry would be useful. We therefore ran the model at 2 mM DIC and 0.5 mM DIC assuming that DIC uptake is split 50% between HCO_3^- uptake and CO_2 uptake. The results indicate that a larger pH increase will be observed at 0.5 mM DIC, suggesting that the experimental data are most likely explained by the decrease in buffer capacity (L266-268). Whilst the model is extremely useful for examining expected trends in the experimental data, there are many assumptions in the model (mode of DIC uptake, rate of photosynthesis, size and shape of cell) that make it tricky to predict precisely what we might observe experimentally. We are therefore cautious about using this estimate to directly gauge whether there has been any reduction in eCA activity in the experimental data.

Follow-up comment: I appreciate the additional model run which is certainly useful, yet I think the discussion of the different DIC treatments still requires some clarification. I assume the low DIC scenario (0.5 mM DIC) yields stronger pH variations because also alkalinity is lower in this scenario compared to the 2 mM standard scenario. This might be meant by 'lower buffer capacity', but it should be clarified in the manuscript to avoid confusion. Note that the decrease in the Revelle factor - which can also be referred to as buffer capacity - at elevated DIC has the opposite effect. This is actually shown in the 'high CO₂ experiment' in Fig. 6 and S6, where the low pH treatment (which must have higher DIC concentrations since it was achieved by bubbling with CO₂ as far as I understand) lead to stronger proton variations, whereas in the example discussed in the response to my question (0.5 mM vs 2 mM DIC), the low DIC treatment lead to stronger proton variations. I think a note on bulk carbonate chemistry (incl. alkalinity) in the different media prepared for the experiment and assumed in the model as well as a definition of the term buffer capacity would help avoid confusion here.

Response: When referring to changes in pH, the term 'buffer capacity' (β) is a measure of the ability of a buffer solution to resist changes in pH. It is sometimes referred to as 'buffer intensity' and is often defined as the amount of acid or alkali addition required to change the pH of the buffer solution (1 L) by one pH unit. The buffer capacity will therefore be reduced by lowering the amount of the buffer in solution or by changes in the initial pH of the buffer away from the pK_a of the buffer, where buffering is greatest. 'Buffer capacity' is distinct from 'total alkalinity', which is a measure of the sum of the bases that are titratable with strong acid.

Therefore, in the low DIC experiment, the buffer capacity is lower at 0.5 mM DIC as DIC is the major contributor to the buffering of the solution. In the high CO₂ experiment, the buffering capacity of the solution is lower at pH 7.6 compared to pH 8.2 because buffering by the HCO₃⁻/CO₃²⁻ equilibrium has a lower effect as the pH moves away from this equilibrium (pK_a 9). Total alkalinity is greatly reduced at 0.5 mM DIC compared to 2 mM DIC, but is not affected by adjusting the pH by CO₂ bubbling.

The Revelle factor refers to CO₂ buffering, rather than changes in pH.

We have clarified these terms in the manuscript to help with the understanding of the changes in cell surface pH (ln 269-270 and ln 319-322).